# Plasticity-induced repression of Irf6 underlies acquired resistance to cancer immunotherapy in pancreatic ductal adenocarcinoma

Il-Kyu Kim [1,2,12], Mark S. Diamond[1,2,12], Salina Yuan[1,2,12], Samantha B. Kemp[1,2], Benjamin M. Kahn[1,2,3], Qinglan Li[4,5], Jeffrey H. Lin [1,2], Jinyang Li[1,2], Robert J. Norgard[1,2], Stacy K. Thomas [1,2], Maria Merolle[1,2], Takeshi Katsuda [1,2], John W. Tobias [6], Timour Baslan[7], Katerina Politi[8,9,10], Robert H. Vonderheide [1,2,11] ✉ & Ben Z. Stanger [1,2,3] ✉

Acquired resistance to immunotherapy remains a critical yet incompletely understood biological mechanism. Here, using a mouse model of pancreatic ductal adenocarcinoma (PDAC) to study tumor relapse following immunotherapy-induced responses, we find that resistance is reproducibly associated with an epithelial-to-mesenchymal transition (EMT), with EMT-transcription factors ZEB1 and SNAIL functioning as master genetic and epigenetic regulators of this effect. Acquired resistance in this model is not due to immunosuppression in the tumor immune microenvironment, disruptions in the antigen presentation machinery, or altered expression of immune checkpoints. Rather, resistance is due to a tumor cell-intrinsic defect in T-cell killing. Molecularly, EMT leads to the epigenetic and transcriptional silencing of interferon regulatory factor 6 (*Irf6*), rendering tumor cells less sensitive to the pro-apoptotic effects of TNF-α. These findings indicate that acquired resistance to immunotherapy may be mediated by programs distinct from those governing primary resistance, including plasticity programs that render tumor cells impervious to T-cell killing.

Immune checkpoint blockade (ICB) has transformed cancer treatment for multiple malignancies[1,2], but durable clinical responses remain elusive in many patients. Beyond the problem of primary resistance (i.e., patients who never respond to ICB), acquired resistance to immunotherapy represents an important clinical challenge. For example, approximately one third of patients with metastatic melanoma who had objective responses to ICB in a recent clinical trial subsequently relapsed over ~2 years of continuous therapy[3]. Similarly, although PD-1 and PD-L1 therapeutic blockade has revolutionized treatment of patients with non-small cell lung cancer (NSCLC), relapse after initial response is a major challenge[4].

Unlike primary resistance, which is attributable to factors present when therapy is initiated, acquired resistance emerges over time. In patients treated with targeted therapies (e.g., EGFR inhibition), acquired resistance is often associated with cellular plasticity, a phenomenon that broadly describes changes in cell identity along a phenotypic spectrum[5]. One of the most well-studied examples of cellular plasticity is epithelial-to-mesenchymal transition (EMT), during which carcinoma cells lose their epithelial features and acquire the more motile characteristics of fibroblasts and leukocytes[6,7]. In addition to its phenotypic effects on tumor cells, EMT has also been associated with the development of an immunosuppressive tumor microenvironment

(TME) in NSCLC, breast cancer, and melanoma[8–11]. These findings have implications for immunotherapy, as exemplified by a recent report that the EMT transcription factor SNAIL promotes primary resistance to ICB in breast carcinomas through its effect on the immunosuppressive CD73/adenosine pathway[12].

Studies of acquired resistance to immunotherapy have been difficult due to a lack of tractable biological systems with which to model tumor relapses over prolonged periods of time. In this work, we develop an immunotherapy-sensitive model of pancreatic ductal adenocarcinoma (PDAC) in which recurrent (and therapy-resistant) disease appears weeks-to-months after a complete response to treatment. Using this system, we delineate the underlying cellular and molecular programs, revealing mechanisms of acquired resistance distinct from those associated with primary non-responsiveness.

## Results

### A model for acquired immunotherapy resistance

Pancreatic cancer is known to respond poorly to ICB. However, a fraction of patients with PDAC clinically respond to combinations including chemotherapy, ICB, and CD40 agonist[13,14], similar to observations in the KPC model system[15–17]. To establish a model of acquired resistance in this setting, we evaluated the efficacy of various chemo- and immuno-therapy drug regimens in subcutaneously implanted 4662 cells[17], a female cell line derived from the KPC mouse model of PDAC (Kras$^{LSL-G12D/+}$; Trp53$^{LSL-R172H/+}$; Pdx1-Cre). We treated 4662 tumor-bearing mice with a single dose of gemcitabine (G) and/or nab-paclitaxel (A), seven doses anti-PD-1 Ab (P) and/or three doses of anti-CTLA-4 Ab (C) every three days, and one dose of agonistic anti-CD40 Ab (F). Neither chemotherapy with gemcitabine and nab-paclitaxel nor ICB with anti-CTLA-4 Ab and anti-PD-1 Ab had significant antitumor effects (Supplementary Fig. 1a), as we previously reported[17,18]. However, combination regimens that included agonistic anti-CD40 Ab and ICB resulted in tumor regressions and prolonged survival, including complete responses (CRs) (Fig. 1a, b and Supplementary Fig. 1a). Mice with CRs following immunotherapy (FCP) showed significant expansion of effector memory (T$_{EM}$) CD8 T cells in the spleen and draining lymph nodes (dLN) (Supplementary Fig. 1b). Accordingly, tumor growth following rechallenge was delayed, an effect that was dependent upon T cells but not NK cells (Supplementary Fig. 1c–e). Thus, the combination of CD40 agonist and ICB leads to durable CRs and T cell memory in this syngeneic model of PDAC.

Despite the potent antitumor activity elicited with combination therapy, responses were heterogenous among treated mice and could be categorized as follows: (i) mice that did not respond initially, reflecting primary resistance; (ii) mice in which therapy delayed tumor growth but did not induce full regression; (iii) mice that fully regressed and then spontaneously relapsed afterwards, reflecting acquired resistance; and (iv) mice exhibiting a durable CR (Fig. 1a–c). Importantly, tumors that recurred after an initial CR or near CR displayed acquired resistance to therapy, since re-treatment with combination therapy yielded no responses (Fig. 1d). We therefore hypothesized that recurrent tumors, particularly those with a late escape phenotype, represented tumors that successfully evaded therapy-induced T cell surveillance and adopted a stable resistant state.

To test this hypothesis, we established tumor cell lines from cohorts of mice that had different therapeutic outcomes following combination therapy. We first compared therapy responses by re-inoculating each tumor cell line into naïve mice and administering combination therapy including gemcitabine, nab-paclitaxel, agonistic anti-CD40 Ab, anti-CTLA-4 Ab, and anti-PD-1 Ab (GAFCP). Cell lines derived from non-responsive tumors (termed "Early Progressor (EP) lines", n = 2) gave rise to tumors exhibiting variable responses to combination therapy, with rates of regression and CR that were comparable to mice bearing control cell lines that were never exposed to therapy (termed "Ctrl lines", n = 4) (Fig. 1e–g). By contrast, cell lines derived from tumors that underwent CR followed by relapse (termed "Escape (Esc) lines", n = 8) gave rise to resistant tumors exhibiting poor survival (Fig. 1e–g). These results suggest that the mechanisms allowing tumors to grow after therapy-induced CR (i.e. acquired resistance) persist in the Esc lines, whereas the mechanisms that render tumors non-responsive upon first exposure to combination therapy (i.e. primary resistance) are not stably preserved in the EP lines.

### EMT confers immunotherapy resistance in PDAC

We considered two models that could explain the emergence of resistant tumors. First, tumor escape might have resulted from the selective expansion of pre-existing resistant subclones. Alternatively, cell plasticity, in the setting of immunotherapy, might have fostered the emergence of newly resistant clones. To distinguish between these models, we generated clonal PDAC lines from the 4662 parental line and examined therapy responses. Consistent with our earlier findings with the 4662 parental line, individual clones exhibited heterogeneous therapeutic outcomes including escape tumors following CR (Supplementary Fig. 1f), suggesting that the resistant phenotype is likely an emergent property of the cancer cells rather than the result of outgrowth of pre-existing clones. As expected, clonal tumors that escaped following an immunotherapy-induced CR (C10.e1 and C7.e1) were highly resistant to combination therapy compared to control tumors (Supplementary Fig. 1g).

Next, we set out to understand the mechanisms underlying acquired resistance. We began by performing bulk RNA sequencing on parental 4662 cells, EP cells, and Esc cells. Principle component analysis (PCA) revealed a strong similarity between parental cells and EP cells, while Esc cells diverged from both, suggesting that Esc cells had acquired a unique transcriptional profile (Fig. 2a). We also observed striking morphological differences; namely, parental cells exhibited epithelial features and gave rise to well-differentiated tumors, while Esc cells exhibited spindle-like features and gave rise to poorly differentiated tumors (Fig. 2b and Supplementary Fig. 1h). In accordance with these observations, gene set enrichment analysis (GSEA) showed that Esc cells were highly enriched for the Hallmark EMT gene signature compared to parental tumors (Fig. 2c) and exhibited a decrease in mRNAs associated with the epithelial phenotype and an increase in mRNAs associated with the mesenchymal phenotype (Supplementary Figs. 1i, 2a). This pattern was present in the original tumor tissues, as Esc tumors showed decreased expression of the epithelial marker E-cadherin and increased expression of the mesenchymal markers Vimentin and Twist (Supplementary Fig. 2b). In addition to EMT, which was the most significantly enriched gene set in Esc tumors, other gene sets enriched in Esc tumors included interferon response, angiogenesis, hypoxia, and inflammation response, while gene sets that were reduced in Esc tumors included androgen/estrogen response and cholesterol homeostasis (Supplementary Fig. 2c). Importantly, Esc lines derived following treatment with immunotherapy alone (FC or FCP), without chemotherapy, also exhibited EMT phenotypes and poor response to immunotherapy (Supplementary Fig. 2d–f), suggesting that these phenotypic changes were a result of immune pressure rather than chemotherapy.

Given the well-documented role of EMT in various forms of therapy resistance[19], we hypothesized that EMT was not merely associated with acquired resistance to immunotherapy but was itself acting as a driver of resistance. To test this, we performed gain-of-function studies to determine which, if any, EMT-TFs contribute to immunotherapy resistance. We found that Zeb1 and Snail family members, but not Twist1, abrogated immunotherapy responses and worsened mouse survival (Supplementary Fig. 3a, b). Thus, we focused on the Zeb1 and Snail EMT-TFs in subsequent studies of immunotherapy resistance. First, we confirmed that overexpression of Zeb1 and Snail in parental tumors, or ablation of both genes in Esc tumors, prompted the expected changes in epithelial-mesenchymal phenotype and

morphology (Supplementary Fig. 3c–f). Next, we performed RNA sequencing and GSEA of the engineered lines. *Zeb1* and *Snail* overexpression in parental tumors (*Zeb1/Snail* OE) resulted in the enrichment of gene signatures associated with Esc tumors in vitro and in vivo (Supplementary Fig. 3g, h), indicating that the transcriptional changes induced by these EMT-TFs resemble those associated with acquired resistance to immunotherapy. Consistent with these findings, *Zeb1/Snail* OE tumors exhibited reduced responses to immunotherapy compared to EV-transduced controls, resulting in poorer survival (Fig. 2d, e). By contrast, ablation of *Zeb1* and *Snail* (*Zeb1⁻/⁻Snail⁻/⁻*) in Esc tumors rescued the response to immunotherapy, leading to greater survival in treated mice (Fig. 2f, g). Importantly, none of the tumor cell lines tested above had a defect in cell growth in vitro (Supplementary Fig. 3i–k).

The immune TME can vary with tissue sites. Therefore, we asked whether EMT would also emerge as a mechanism of acquired resistance in tumors that respond and relapse following immunotherapy in the pancreas rather than the subcutaneous microenvironment. To this end, we orthotopically transplanted parental 4662 tumor cells and monitored tumor growth by ultrasound screening upon immunotherapy. Consistent with the subcutaneous tumor model, orthotopic Esc tumors that relapsed after CR or near CR (8-12 weeks post implantation) were poorly differentiated and exhibited a mesenchymal morphology, whereas control tumors maintained their epithelial

features (Fig. 2h). Accordingly, resistant cells lacked E-cad expression and exhibited elevated *Zeb1* and *Snail* expression (Fig. 2i). Next, we transplanted EV and *Zeb1/Snail* OE tumors into the pancreas and evaluated the response to immunotherapy. Similar to our findings in subcutaneous models, EV tumors showed robust tumor regressions, whereas *Zeb1/Snail* OE tumors exhibited delayed but progressive growth following immunotherapy (Fig. 2j–l). Taken together, these data support the hypothesis that EMT promotes acquired resistance to immunotherapy.

## EMT drives tumor cell-intrinsic resistance to cytotoxic T cell activity

Resistance to immunotherapy in various PDAC models has been associated with an immunosuppressive TME characterized by abundant granulocytic myeloid-derived suppressor cells (gMDSCs) and a paucity of dendritic cells (DCs) and CD8 T cells[16,20,21]. To determine whether EMT fostered the creation of an immunosuppressive TME, we compared the immune profiles associated with parental and EV controls vs. Esc and *Zeb1/Snail* OE tumors. Contrary to expectations, both Esc and *Zeb1/Snail* OE tumors exhibited decreased infiltration of gMDSCs and increased infiltration of CD103⁺ DCs (cDC1) and CD8⁺ T cells compared to parental tumors (Fig. 3a–c). Similar changes were observed in the original Esc tumors, which consistently accumulated fewer Ly6G⁺ myeloid cells and more CD8a⁺ lymphocytes than 4662

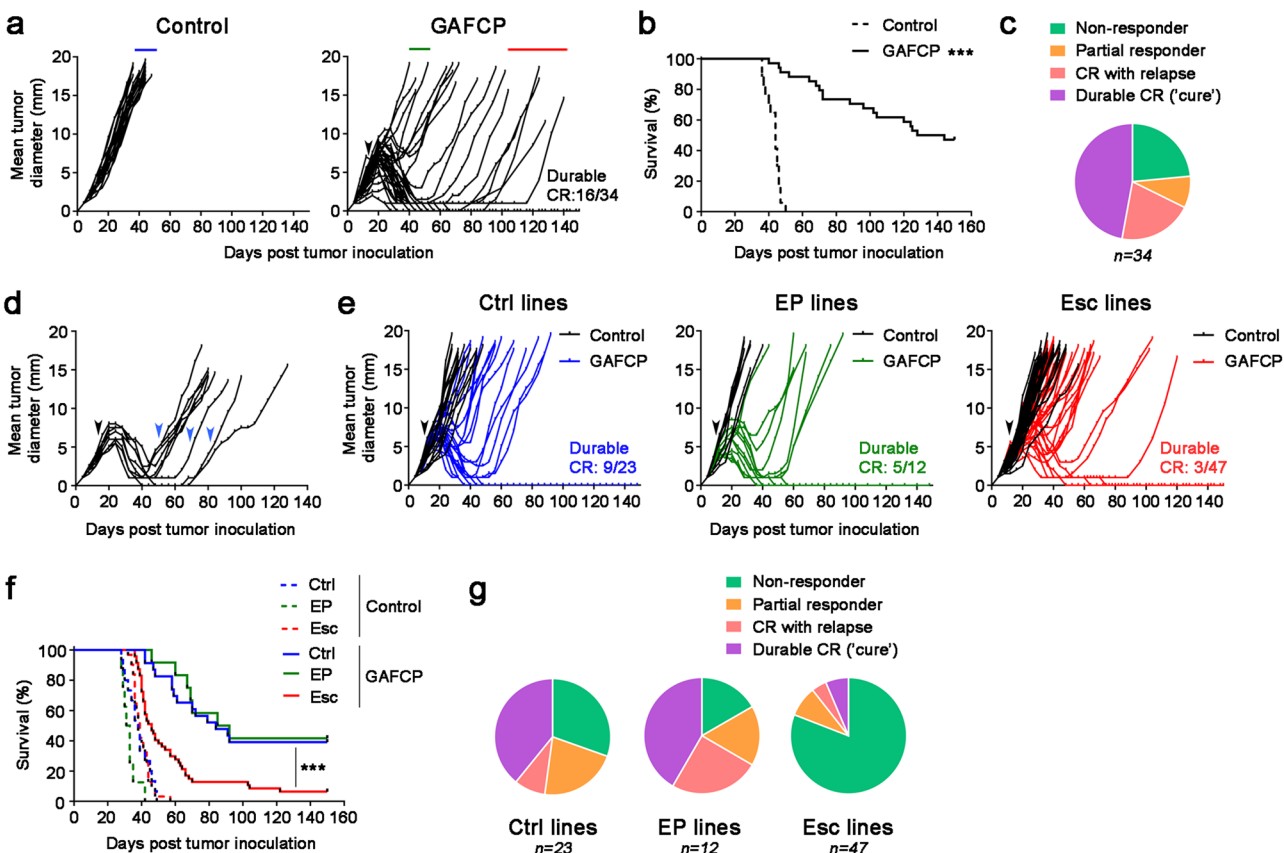

**Fig. 1 | Recurrent PDAC acquires resistance to combination chemoimmunotherapy. a, b** Mice were injected subcutaneously (s.c.) with 4662 PDAC cells and treated intraperitoneally (i.p.) with control IgG (*n* = 17) or chemoimmunotherapy (*n* = 34) (black arrow) consisting of gemcitabine (G), nab-paclitaxel (A), αCD40 agonistic Ab (F), αCTLA-4 Ab (C), and αPD-1 Ab (P). Tumor growth (**a**) and survival (**b**) were monitored. **c** The proportion of treated mice exhibiting no response/early progression, partial response, relapse after complete response (CR), or durable CR is depicted. **d** Mice with recurrent tumors after CR or near CR were re-treated with GFCP (blue arrows) and tumor size measured (*n* = 9). **e, f** Tumor cell lines were

generated from s.c. tumors treated with control IgG ('Ctrl' lines, *n* = 4) or from tumors exhibiting early progression ('EP' lines, *n* = 2) or relapse after CR ('Esc' lines, *n* = 8) on chemoimmunotherapy (denoted by blue, green, and red lines in **a**, respectively). Naïve WT mice were challenged s.c. with these cell lines, and the resulting tumors were treated with control IgG (*n* = 3 or 4 per line) or GAFCP (*n* = 5 or 6 per line) (black arrow). Tumor growth (**e**) and survival (**f**) were monitored. **g** Response rates following treatment for each class of tumor cell line in **e, f** are shown. *** *P* < 0.0001 by log-rank (Mantel-Cox) test (**b, f**). Source data are provided as a Source Data file.

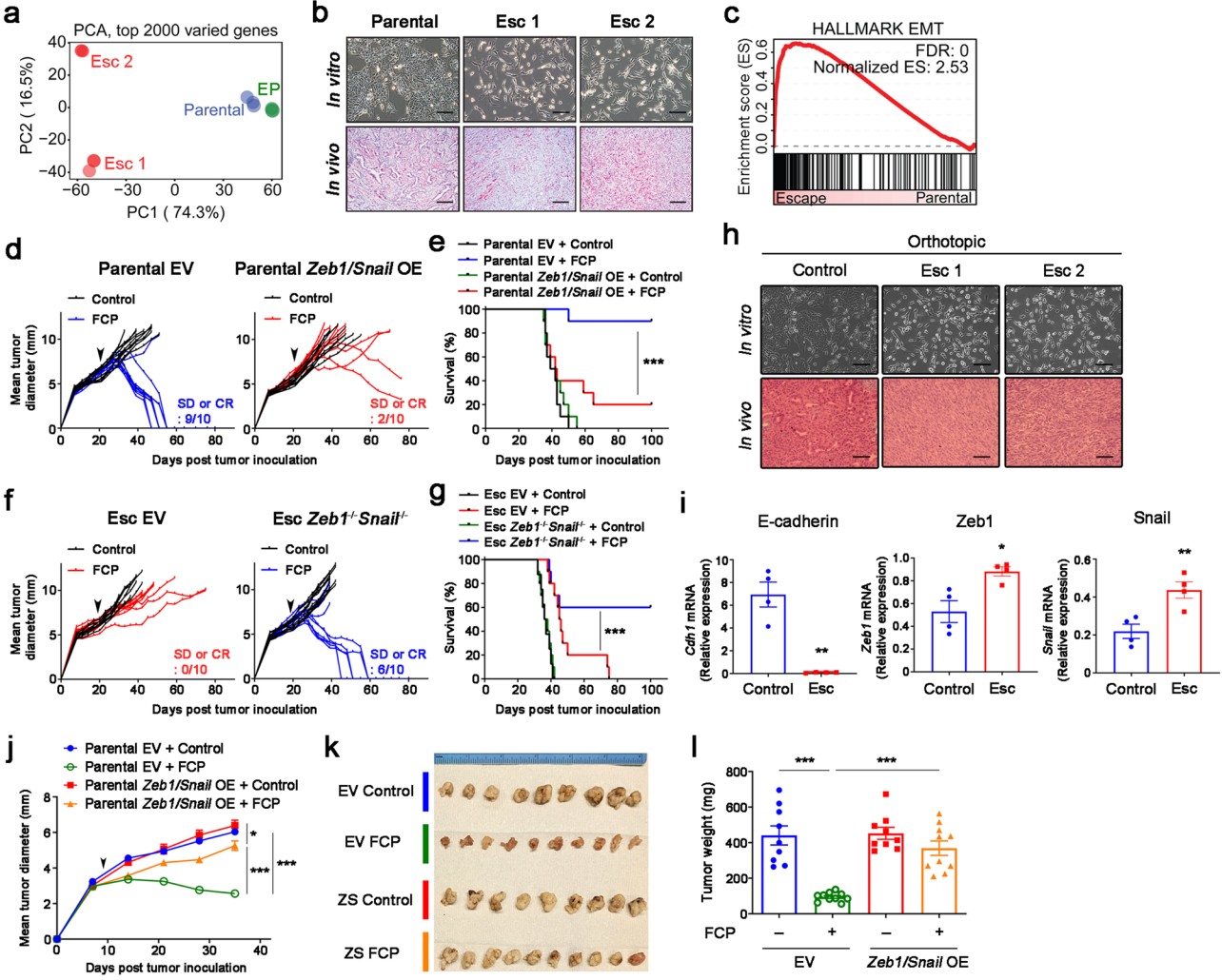

**Fig. 2 | EMT induces immunotherapy resistance in PDAC. a** PCA plot of RNA-seq data from parental, EP, and Esc cell lines (triplicates). **b** Representative bright field (top) and H&E (bottom) images of cultured cells and s.c. implanted tumors on day 18. Scale bars, 250 μm. **c** GSEA of the EMT Hallmark (Molecular Signature Database) in 4662 parental vs. Esc cell lines. Normalized enrichment score (NES) and false discovery rate (FDR) are shown. Individual tumor growth (**d**) and survival (**e**) of mice bearing s.c. implanted 4662 parental empty vector (EV, left) and *Zeb1/Snail* OE (right) tumors treated with either control IgG or FCP (arrow) (*n* = 10). Individual tumor growth (**f**) and survival (**g**) of mice bearing s.c. implanted 4662 Esc EV (left) and *Zeb1−/−Snail−/−* (right) tumors treated with either control IgG or FCP (arrow) (*n* = 10). SD, stable disease. Clonal 4662 (C7 and C10) and derived Esc (C7.e1 and C10.e1) lines were used for genetic modification, and both lines showed a similar

phenotype. **h** Representative bright field (top) and H&E (bottom) images of in vitro and in vivo (orthotopic) control and Esc tumors following treatment with control IgG and immunotherapy (FCP), respectively. Scale bars, 250 μm. **i** qPCR of *E-cad*, *Zeb1*, and *Snail* in control and Esc tumor cell lines established from orthotopic tumors (*n* = 4). Mean tumor diameter over time by ultrasound imaging of orthotopic 4662 EV or *Zeb1/Snail* OE tumors treated with control IgG or FCP (arrow) (**j**). Bright field dissection images (**k**) and tumor weights (**l**) of orthotopic pancreas tumors 6 weeks post tumor implantation (*n* = 9 for controls and 10 for FCP). Data are presented as mean ± SEM. *$P < 0.05$, **$P < 0.01$, ***$P < 0.0001$ by log-rank (Mantel-Cox) test (**e**, **g**), Student's *t* test (**i**), and one-way ANOVA (**j**, **l**). Data represent two independent experiments. Source data and exact *P* value are provided as a Source Data file.

control and EP tumor tissues (Supplementary Fig. 4a, b). Similar results were also observed when these comparisons were made in the orthotopic setting (Supplementary Fig. 5a, b), suggesting that EMT has comparable effects on the TME regardless of tumor sites. Also unexpectedly, the expression of co-inhibitory molecules such as PD-L1 and CD73 was reduced in both Esc and *Zeb1/Snail* OE tumors, although the expression of the TIGIT ligand CD155 was slightly increased compared to parental controls (Supplementary Fig. 5c, d). Conversely, ablation of *Zeb1* and *Snail* in Esc tumors resulted in a paradoxical increase in immunosuppressive gMDSCs and a decrease in total T cells (Supplementary Fig. 5e).

Given these surprising findings, we sought to understand how EMT directs the formation of immune-favorable TMEs. For this, we performed cytokine/chemokine arrays using culture supernatants from EV and *Zeb1/Snail* OE tumors. In addition, we assessed the

abundance of transcripts for specific cytokines/chemokines in Esc and *Zeb1/Snail* OE tumors. Esc and *Zeb1/Snail* OE tumors expressed lower levels of gMDSC-recruiting cytokines and chemokines than parental and EV tumor cells, including G-CSF, GM-CSF, IL-1a, and CXCL2 (Supplementary Fig. 6a–c). Consistent with increased M-CSF expression, Esc and *Zeb1/Snail* OE tumors exhibited an increased abundance of tumor-associated macrophages (TAMs) (Supplementary Fig. 6d, e). While the frequency of Arginase I+ TAMs was elevated in these resistant tumors compared to parental and EV tumors, most TAMs were MHC II+ M1 macrophages with high expression of CD80 and CD86 (Supplementary Fig. 6d, e). Accordingly, CD8 T cells in Esc and *Zeb1/Snail* OE tumors showed intact or even greater activation and cytolytic molecule expression than those in parental and EV tumors, respectively (Supplementary Fig. 6f, g). Taken together, these results suggest that EMT promotes resistance to

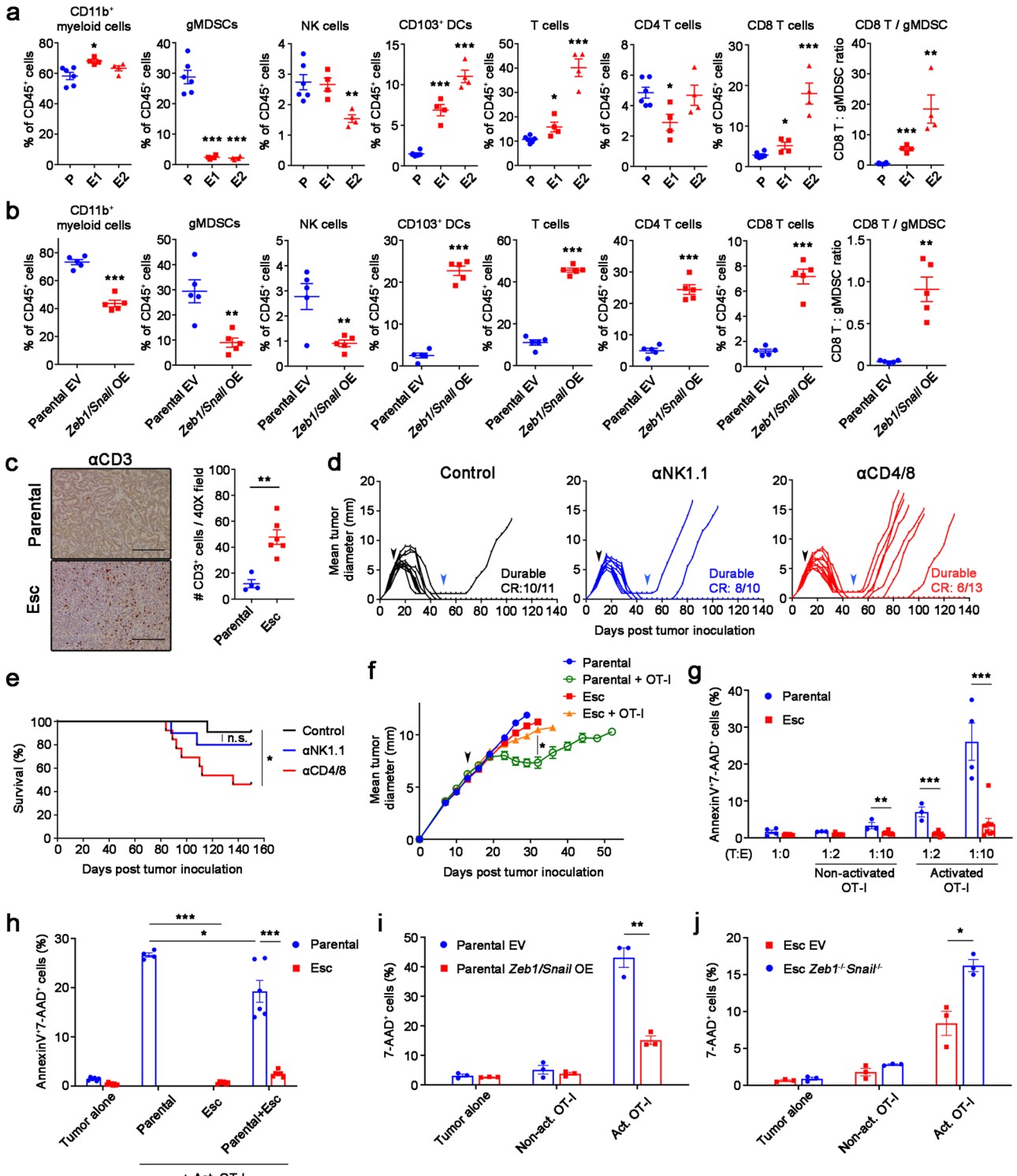

immunotherapy by mechanisms other than the creation of an immunosuppressive TME.

To determine whether loss of MHC I and/or Ag presentation accounted for acquired resistance in our model, we transduced parental and Esc tumor cells with chicken *ovalbumin* (OVA) and assessed the relative intensity of antigen presentation using an antibody (Anti-H-2K$^b$ bound to SIINFEKL) that detects OVA peptide in the context of MHC class I. Although Esc lines exhibited mild heterogeneity in antigen presentation (some slightly increased and some slightly decreased compared to parental; Supplementary Fig. 7a), all OVA-expressing Esc lines exhibited marked resistance to killing when co-cultured with

OVA-specific CD8$^+$ T cells (OT-I) (Supplementary Fig. 7b). In agreement with this result, Esc and *Zeb1/Snail* OE tumors in vivo had comparable MHC I expression to parental control tumors (Supplementary Fig. 7c, d). These results suggest that mechanisms other than MHC I down-regulation account for EMT-associated acquired resistance in our model.

T cells play a crucial role in limiting tumor relapse following tumor clearance[22–24]. We confirmed in our model that animals achieving immunotherapy-induced CR required T cells, but not NK cells, to maintain tumor control (Fig. 3d, e). This suggested two possibilities: (i) EMT induces a tumor cell intrinsic resistance to T cell killing, and/or (ii)

**Fig. 3 | EMT mediates cell-autonomous resistance to direct cytolytic T cell killing.** Flow cytometric analysis of immune populations in s.c. implanted 4662 parental (P, $n = 6$) vs. Esc (E1 and E2, $n = 4$ per line) (**a**) and 4662 parental EV ($n = 5$) vs. *Zeb1/Snail* OE ($n = 5$) tumors (**b**) on day 18 post inoculation. **c** Representative αCD3 IHC images (left) and quantitation (right) from s.c. implanted parental and Esc tumors ($n = 4$ and 6 respectively). Scale bars, 250 μm. Mice that had been transplanted with 4662 parental cells and achieved CR or near CR following combination therapy (black arrows) were treated with control IgG ($n = 11$), depleting αNK1.1 Ab ($n = 10$), or αCD4 and αCD8 Abs ($n = 13$), starting from day 50 (blue arrows) post tumor inoculation, and monitored for tumor recurrence (**d**). The corresponding survival curves are shown in (**e**). n.s., non-significant. **f** Tumor growth of s.c. inoculated OVA-transduced 4662 parental and Esc tumors in NOD/SCID mice, with or without adoptive transfer of activated OVA-specific CD8$^+$ T cells (OT-I) on day 14 (arrow) ($n = 6$ for parental and 8 for the other groups). **g** OVA-

tdTomato$^+$ 4662 parental and Esc tumors were co-cultured with non-activated or activated OT-I by αCD3 and αCD28 Abs overnight, at indicated tumor to effector (T:E) ratios. Each dot represents biological replicates. Two days later, AnnexinV and 7-AAD expression on tumor cells were determined by flow cytometry. **h** OVA$^+$ 4662 parental and Esc tumors were additionally transduced with YFP or CFP expression constructs, plated separately or mixed, and co-cultured with activated OT-I. AnnexinV and 7-AAD on each tumor were measured 2 d after co-culture. The percentages of 7-AAD$^+$ cells on co-cultured OVA$^+$ 4662 parental EV vs. *Zeb1/Snail* OE tumors (**i**) and 4662 Esc EV vs. *Zeb1$^{-/-}$/Snail$^{-/-}$* tumors (**j**) with or without activated OT-I for 2 d ($n = 3$). Data are presented as mean ± SEM. \*$P < 0.05$, \*\*$P < 0.01$, \*\*\*$P < 0.0001$ by Student's $t$ test (**a**, **b**, **g**–**j**), Mann-Whitney $t$ test (**c**), log-rank (Mantel-Cox) test (**e**), and one-way ANOVA (**f**). Data represent two independent experiments. Source data and exact $P$ value are provided as a Source Data file.

EMT in tumor cells induces a defect in T cell function. To explore these possibilities, we transplanted OVA-expressing parental and Esc tumor cells into NOD/SCID mice and measured tumor growth before and after infusion with activated OT-I T cells. OVA-expressing parental tumors responded to the OT-I infusion with slowing of tumor growth (Fig. 3f). However, OVA-expressing Esc tumors grew at a similar rate as control tumors despite the presence of OT-I T cells (Fig. 3f). These results suggest that tumors escape immune surveillance by evading antigen-specific T cell immunity.

Next, we established an in vitro co-culture system to determine whether the mechanism of resistance involves a direct interaction between tumor cells and CD8 T cells. Whereas parental cells were killed in the presence of either activated or non-activated OT-I cells (measured as positivity for AnnexinV and 7-AAD), Esc cells were highly resistant to OT-I killing (Fig. 3g). Next, to determine whether resistance to killing was cell-autonomous, we labelled parental and Esc tumors with different fluorescent markers and co-cultured them individually or together with OT-I cells. In this mixed co-culture, Esc cells were far more resistant to killing than parental cells (Fig. 3h), suggesting that a tumor cell-intrinsic block to T cell killing drives the resistance phenotype. In addition, we found that naïve OT-I cells were poorly primed when co-cultured with Esc cells compared to parental cells (Supplementary Fig. 7e, f). Consistent with this finding, we noted that parental cells mixed with Esc cells had reduced cell death in OT-I co-culture compared to parental cells co-cultured alone (Fig. 3h; compare "Parental" to "Parental +Esc"). Taken together, these results indicate that Esc cells are intrinsically resistant to T cell killing and have a mild defect in T cell priming ability.

Because resistance arises through epithelial-mesenchymal plasticity rather than outgrowth of pre-existing resistant subclones (Supplementary Fig. 1f), we next studied whether the immune pressure present in the in vitro OT-I co-culture system might induce a similar change. Remarkably, parental cells that survived 2 days of co-culture exhibited reduced expression of the epithelial marker E-cadherin, suggestive of an EMT-like process (Supplementary Fig. 8a). To determine whether immunotherapy prompts a similar shift in epithelial-mesenchymal phenotypes in vivo, we implanted parental tumor cells and compared the transcriptional profiles of treated tumors to those of control (untreated) tumors. GSEA revealed that even this short-term immunotherapy (FCP) caused tumor cells in vivo to become enriched for signatures associated with the Esc tumors and the Hallmark EMT signature (Supplementary Fig. 8b, c). Together, these findings strongly suggest that immune pressure promotes the expansion of PDAC cells with a more mesenchymal phenotype that confers resistance to T cell killing. In line with this idea, we found that *Zeb1/Snail* OE rendered parental tumor cells resistant to killing by OT-I cells (Fig. 3i), whereas depletion of *Zeb1* and *Snail* made Esc tumor cells more sensitive to T-cell killing (Fig. 3j).

## Transcriptional and epigenetic regulation of Irf6 contributes to acquired immunotherapy resistance

Given the stability of the EMT-associated resistance phenotype, we reasoned that the underlying mechanism was likely to involve epigenetic remodeling. Consequently, we performed ATAC-seq on parental EV and *Zeb1/Snail* OE tumors to identify genes whose chromatin accessibility changed upon EMT induction (in both steady state and in co-culture with OT-I cells) (Fig. 4a). In parallel, we examined the overlap of EMT-associated transcriptional differences across two experimental comparisons – (i) parental cells vs. Esc cells and (ii) EV- vs. *Zeb1/Snail*-transduced parental cells – and then used GSEA to compile a list of candidate genes whose transcriptional regulation correlated with immune sensitivity across both datasets (Supplementary Table 1). An examination of these epigenetically and transcriptionally regulated gene lists yielded a single gene candidate common to both: interferon regulatory factor 6 (*Irf6*).

Based on these findings, we hypothesized that Irf6 plays a role in EMT-associated resistance to T cell killing. As predicted bioinformatically, *Zeb1/Snail* OE resulted in a loss of chromatin accessibility of the *Irf6* locus (Fig. 4b), particularly at the promoter region (Fig. 4c and Supplementary Fig. 9a, b), leading to a corresponding decrease in *Irf6* mRNA (Fig. 4d). In agreement with this finding, *Zeb1/Snail* OE also resulted in the downregulation of putative Irf6 target genes but not those of unrelated transcription factors such as *Six2* (Fig. 4e and Supplementary Fig. 9c–e). In addition, we found that ZEB1 and SNAIL directly bound to the promoter region of *Irf6* by ChIP-qPCR assay (Fig. 4f). Although IRF6 was readily detected by immunostaining of the original control and EP tumors, IRF6 was barely detectable in the original Esc tumors (Fig. 4g), demonstrating that loss of IRF6 is associated with acquired resistance to immunotherapy. Next, using gene sets generated via ectopic expression of *Irf6* in PDAC cells, we found that *Irf6* signatures were strongly enriched in therapy-sensitive parental and EV tumors compared to Esc and *Zeb1/Snail* OE tumors, respectively (Fig. 4h). Interestingly, analysis of published human scRNA-Seq PDAC datasets revealed that *IRF6* expression is largely restricted to epithelial cells, in contrast to other *IRF* genes[25–27] (Supplementary Fig. 9f–h). Furthermore, *IRF6* signatures from human PDAC were consistently enriched in therapy-sensitive parental and EV tumors as compared to therapy-resistant Esc and *Zeb1/Snail* OE tumors, respectively (Fig. 4i). Taken together, these results nominate Irf6 as a candidate EMT-related driver of immune sensitivity whose loss represents a potential mechanism of acquired resistance to immunotherapy.

## Irf6 restoration promotes cytotoxic T cell killing and response to immunotherapy in resistant PDAC

To functionally assess the role of Irf6 in acquired immunotherapy resistance, we restored *Irf6* expression to Esc tumors and assessed vulnerability to T cell killing in vitro. Esc tumor cells engineered to re-express *Irf6* regained sensitivity to T cell killing upon OT-I co-culture (Fig. 5a), whereas parental cells lacking Irf6 (*Irf6$^{-/-}$*) became resistant to

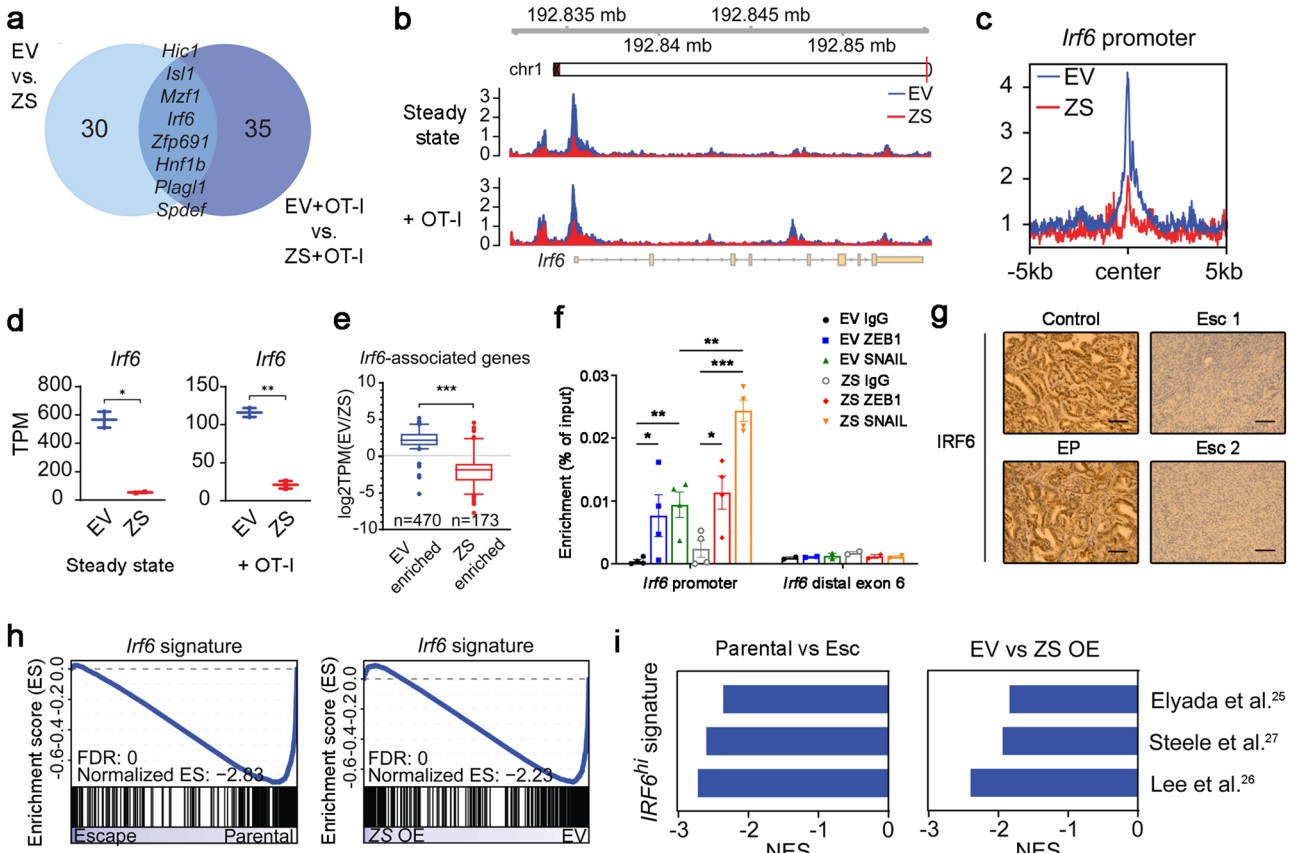

**Fig. 4 | Transcriptional and chromatin profiling identifies Irf6 as a potential regulator of acquired immunotherapy resistance. a** Venn diagram of HOMER de novo motifs identified in chromatin regions significantly enriched in EV vs. *Zeb1/Snail* OE (ZS) cell lines that have (right) or have not (left) been co-cultured with OT-1 cells (duplicates). Significantly enriched chromatin regions were defined as |log2-fold change| > 1.4 and *p* value < 0.05 after DESeq2 analysis. **b** Genome browser track showing ATAC-seq reads along the *Irf6* gene, comparing EV vs. *Zeb1/Snail* OE cell lines. **c** Aggregate plots comparing the average ATAC signal of EV (blue) and *Zeb1/Snail* OE (red) tumors around putative *Irf6* promoter sequences. For details, see **Methods**. **d** Transcripts per million (TPMs) of *Irf6* in EV vs. *Zeb1/Snail* OE tumors that have (right) or have not (left) been co-cultured with OT-1 cells (duplicates). **e** Boxplots (log2fold) showing the expression of *Irf6*-associated genes with differentially open chromatin in parental EV (left, *n* = 470 genes) vs. *Zeb1/Snail* OE (right, *n* = 173 genes) tumors. Centre and whiskers represent means and 2.5-97.5 percentiles. **f** ChIP with control IgG or antibodies to ZEB1 and SNAIL in DNA from 4662

parental EV and *Zeb1/Snail* OE (ZS) tumor cells, followed by qPCR quantification in enriched DNA using primers for *Irf6* putative promoter and distal exon 6 regions. Data represent two independent experiments. **g** Representative IHC images of IRF6 in the original primary tumor tissues from control, EP, and Esc tumor-bearing mice. Scale bars, 250 μm. **h** GSEA plots of an *Irf6*-dependent gene signature (derived by comparing *Irf6*-expressing tumors to controls; provided in Supplementary Data 2) in parental vs. Esc (left) (triplicates) and parental EV vs. *Zeb1/Snail* OE (right) (duplicates) tumors. **i** GSEA of gene signatures derived from human PDAC cells that highly express *IRF6* in parental vs. Esc (left) and EV vs *Zeb1/Snail* OE tumors (right). Negative normalized enrichment scores (NES) demonstrate enrichment in parental and EV tumors compared to Esc and *Zeb1/Snail* OE tumors, respectively. Data are presented as mean ± SEM unless otherwise indicated. *$P < 0.05$, **$P < 0.01$, ***$P < 0.0001$ by Student's *t* test (**d**, **e**) and one-way ANOVA (**f**). Source data and exact *P* value are provided as a Source Data file.

---

OT-I cell killing (Fig. 5b). Of note, although *Irf6*-expressing Esc cells maintained their mesenchymal morphology and had similar growth kinetics in culture (Supplementary Fig. 10a, b), epithelial genes such as Ecad, Ocln, and Cldn7 were upregulated compared to control (EV) cells (Supplementary Fig. 10c). Next, we tested the ability of *Irf6* expression to restore immunotherapy responsiveness in vivo. Esc tumors expressing *Irf6* partially recovered their response to immunotherapy compared to control (EV) tumors, with some mice exhibiting durable CR and prolonged survival (Fig. 5c, d). Similarly, *Irf6*-expressing Esc tumors showed improved responses to combination chemoimmunotherapy (Supplementary Fig. 10d, e). By contrast, *Irf6* knockout in parental tumors led to resistance to immunotherapy and worse overall survival (Fig. 5e, f). These results suggest that loss of *Irf6* in association with EMT promotes resistance to T cell killing in vitro and attenuated responses to immunotherapy in vivo, phenotypes that can be rescued by restoration of *Irf6* expression.

To determine whether Irf6 might be associated with acquired resistance to immunotherapy in patients, we scanned the literature for

clinical trials involving immunotherapy in which matched pre- and post-resistance transcriptomes were available for analysis. We succeeded in finding a single dataset meeting this criteria: a study of patients with non-small cell lung cancer who initially responded to ICB (anti-PD-1/PD-L1±anti-CTLA-4 Ab) but later developed resistance[28]. Roughly half of the patients with acquired resistance exhibited decreased expression of *IRF6* in paired comparisons between pre- and post-treatment; in those patients, *Irf6* signatures were enriched in pre-treatment samples (pre-ICB) compared to post-treatment samples with acquired immunotherapy resistance (IR) (Fig. 5g, h). Importantly, EMT signatures were inversely correlated with *IRF6* expression, such that therapy-resistant patients with decreased *IRF6* expression were enriched for EMT signatures compared to pre-ICB (Fig. 5i). By contrast, therapy-resistant patients with no change or an increase in *IRF6* expression showed the opposite result (Fig. 5i). Thus, immunotherapy resistance in a subset of lung cancer patients is associated with loss of *IRF6* expression and concomitant acquisition of an EMT signature.

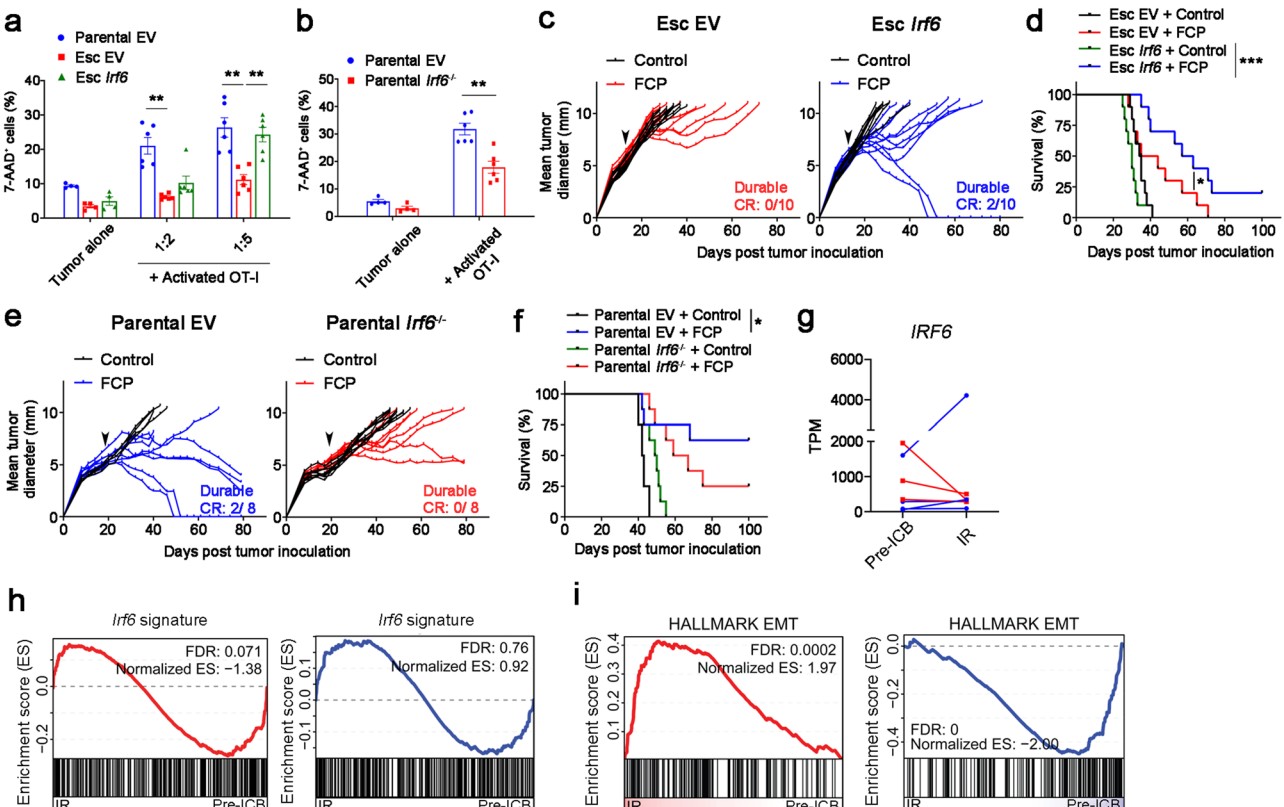

**Fig. 5 | Irf6 loss contributes to EMT-induced immunotherapy resistance. a** OVA-tdTomato⁺ 4662 parental EV, Esc EV, and Esc *Irf6* tumors were co-cultured with or without activated OT-I at the indicated 4662 tumor cell-to-OT-I ratios for 2 d. Tumor cell expression of 7-AAD expression was measured by flow cytometry. **b** OVA-tdTomato⁺ 4662 parental EV and *Irf6* KO tumors were used as target cells (1:5 ratio). For (**a**, **b**) *n* = 4 for tumor alone and 6 for co-cultures. Tumor growth (**c**) and survival (**d**) of mice bearing 4662 Esc EV and Esc *Irf6* tumors treated with control IgG or FCP (arrow) (*n* = 10). Data represent two independent experiments. Tumor growth (**e**) and survival (**f**) of mice bearing 4662 parental EV and parental *Irf6* KO tumors treated with control IgG or FCP (arrow) (*n* = 8). **g** TPMs of *IRF6* in 7 treatment-paired NSCLC patient samples (ref. 28). All patients demonstrated initial response to combinatorial checkpoint blockade (anti-PD-1/PD-L1±anti-CTLA-4 Ab) before relapsing. Lines are drawn from patient-matched early treatment (Pre-ICB) to immunotherapy recurrence (IR). Red lines indicate 3 patient samples demonstrating a decrease in *IRF6* expression with recurrence. Blue lines indicate the others demonstrating unchanged or increased *IRF6* expression with recurrence. GSEA plots of an *Irf6*-dependent gene signature derived from *Irf6*-expressing 4662 tumors (**h**) and the EMT Hallmark (**i**) in patient-matched Pre-ICB vs. IR samples separately assorted based on *IRF6* expression as in (**g**). Data are presented as mean ± SEM. *P < 0.05, **P < 0.01, ***P < 0.0001 by Student's *t* test (**a**, **b**) and log-rank (Mantel-Cox) test (**d**, **f**). Source data and exact *P* value are provided as a Source Data file.

## Irf6 facilitates T cell-mediated tumor control via TNF-induced apoptosis

To understand how Irf6 regulates tumor cell-intrinsic resistance to T cell killing, we compared the transcriptomes of control (EV) and *Irf6*-expressing tumor cells after OT-I co-culture. GSEA identified various hallmark gene sets as enriched (cholesterol homeostasis, MYC targets, estrogen response, TNF-α signaling via NFκB, etc.) or depleted (IFN response and EMT) following ectopic expression of *Irf6* (Supplementary Fig. 10f). Given the known role of TNF and NF-κB in T cell-mediated cell death[29–31], we hypothesized that *Irf6* loss confers resistance to T cell killing by blocking the pro-apoptotic effects of TNF-α. Consistent with this hypothesis, we found that Esc cells were markedly resistant to TNF-α-induced cell death compared to parental cells and *Irf6* re-expression restored sensitivity to TNFα-induced killing (Fig. 6a). Re-expression of *Irf6* in Esc cells had no detectable effect on NF-κB pathway components (Supplementary Fig. 10g). TNF-α-induced killing was due to apoptosis, as *Irf6*-expressing cells exhibited greater staining for cleaved caspase-3 compared to control (EV) Esc cells, both in vitro following TNF-α treatment (Fig. 6b and Supplementary Fig. 11a) and in vivo following immunotherapy (Fig. 6c and Supplementary Fig. 11b). Treatment with the pan-caspase inhibitor z-VAD reversed the enhanced sensitivity to TNF-α in *Irf6*-expressing cells (Fig. 6d), and deletion of *Tradd*, *Fadd*, or *Casp8* – genes encoding intracellular

mediators of TNF-induced cell death – had similar effects (Fig. 6e). *IRF6*-related disorders in humans, including Van Der Woude syndrome, have been linked to point mutations in the *IRF6* DNA binding domain[32]. These mutations introduced into the mouse *Irf6* gene either partially or completely abrogated Irf6's ability to sensitize cells to TNF-α-induced killing (Supplementary Fig. 10h), indicating a role for DNA binding in Irf6's effects. Together, these results suggest that Irf6 confers sensitivity to TNF-induced cell death through a classical TRADD-FADD-CASP8 death receptor signaling pathway.

T cells employ multiple redundant mechanisms to kill their targets. To confirm that death receptor signaling is critical for T cell cytotoxicity of PDAC cells, we assessed the consequences of *Tradd*, *Fadd*, or *Casp8* deletion on T cell-mediated cytotoxicity. Whereas *Irf6* expression restored the ability of OT-I cells to kill OVA-expressing Esc cells, loss of any of these apoptosis mediators significantly blunted the effect (Fig. 6f), suggesting that this pathway plays an important role in T cell killing of these PDAC cells. Next, we used TNF-α neutralizing Ab to determine whether the dependency on Irf6 for efficient T cell killing was specific for TNF-α. Whereas anti-TNF-α antibodies had no effect on T cell-mediated killing of control Esc cells in OT-I co-culture, antibody treatment reduced the killing of *Irf6*-expressing cells to the level of control Esc cells (Fig. 6g). Thus, Irf6 sensitizes PDAC cells to T cell-mediated apoptosis by altering the cellular response to TNF. Finally,

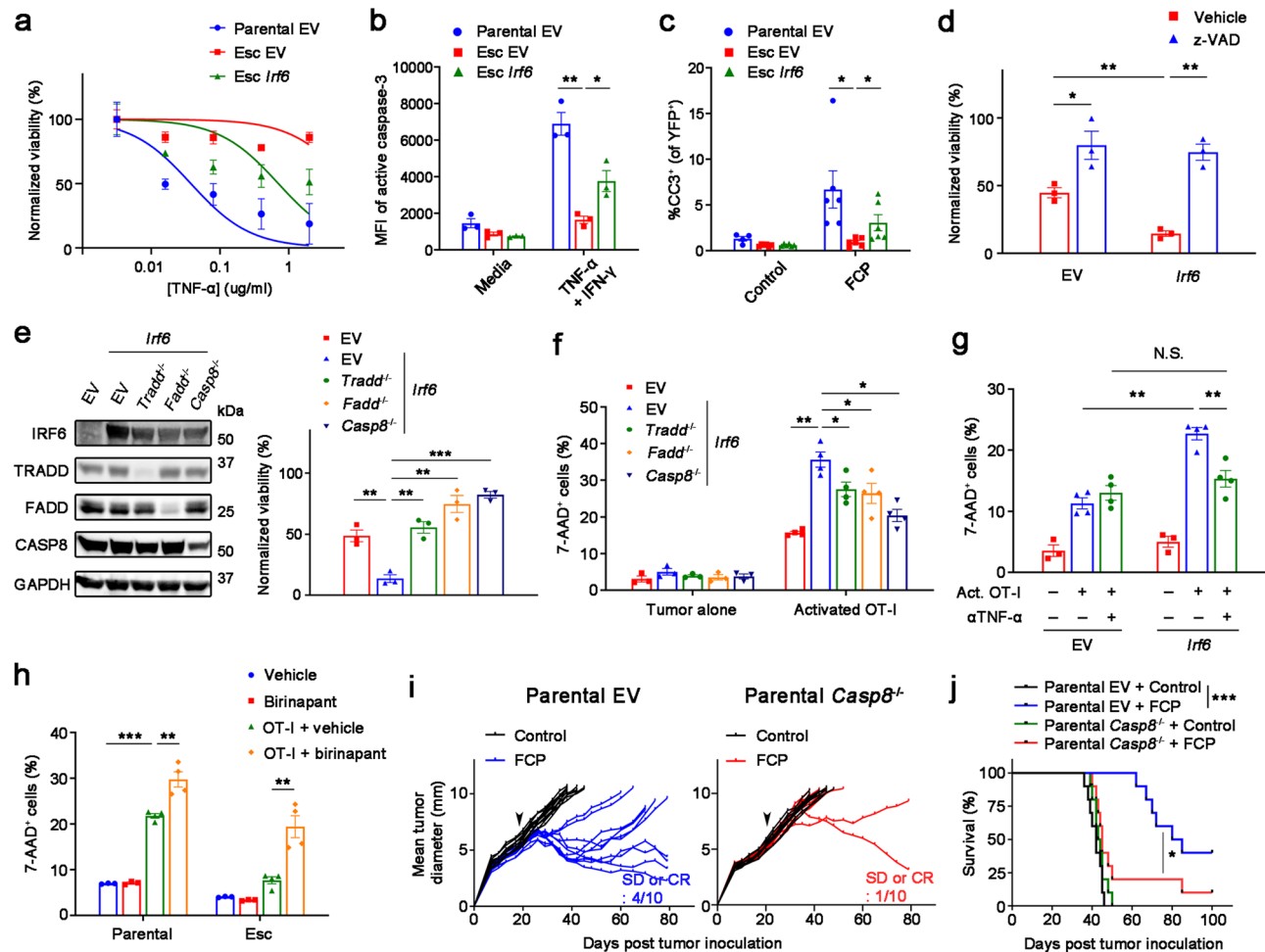

**Fig. 6 | Irf6 promotes susceptibility to T cell killing by enhancing TNF-induced apoptosis. a** Normalized viability of 4662 parental EV, Esc EV, and Esc *Irf6* tumors with varying concentrations of TNF-α in the presence of IFN-γ (0.2 μg/ml) plus cycloheximide (1 μg/ml) for 48 h (*n* = 4). IC₅₀ values are 0.03472 ug/ml for parental EV, 0.6494 ug/ml for Esc *Irf6*, and not determined for Esc EV tumors. Mean fluorescence intensities (MFIs) or percentages of cleaved caspase-3 in 4662 parental EV, Esc EV, and Esc *Irf6* tumors treated with or without TNF-α (0.5 μg/ml) plus IFN-γ in the presence of cycloheximide by flow cytometry (*n* = 3) (**b**) or in s.c. implanted YFP⁺ those tumors with or without immunotherapy by IF staining. Each dot represents biological replicates (**c**). **d** Normalized viability of 4662 Esc EV and Esc *Irf6* tumors, treated with vehicle or z-VAD (20 μM), in response to TNF-α plus IFN-γ in the presence of cycloheximide (*n* = 3). **e** Left, immunoblots of IRF6 and TNF-related cell death mediators in 4662 Esc EV and Esc *Irf6* cells with or without ablation of each gene. Right, normalized viability of 4662 Esc EV and Esc *Irf6* tumors with or without indicated gene ablation in response to TNF-α plus IFN-γ in the presence of cycloheximide (*n* = 3). The percentages of 7-AAD⁺ cells among OVA-tdTomato⁺ 4662 Esc EV and Esc *Irf6* cells with or without the indicated gene ablation (**f**), or the same cells with or without TNF-α neutralizing Ab (5 μg/ml) (**g**), or OVA-tdTomato⁺ 4662 parental and Esc cells in the presence of vehicle or birinapant (5 μM) (**h**), co-cultured with or without activated OT-I for 2 d. *n* = 3 for tumor alone and 4 for co-cultures. Tumor growth (**i**) and survival (**j**) of mice bearing 4662 parental EV and *Casp8* KO tumors treated with control IgG or FCP (arrow) (*n* = 10). Data are presented as mean ± SEM. **P* < 0.05, ***P* < 0.01, ****P* < 0.0001 by Student's *t* test (**b**–**d**), one-way ANOVA (**e**–**h**), and log-rank (Mantel-Cox) **t**est (**j**). Data represent two independent experiments. Source data and exact *P* value are provided as a Source Data file.

we tested whether activating a TNF-induced cell death pathway could sensitize resistant tumors to T cell killing. Indeed, we observed a significant increase in T cell killing of both parental and Esc tumors by adding birinapant, a pharmacological inhibitor of cIAP1/2 facilitating TNF-induced apoptosis[31,33], to the T cell co-culture (Fig. 6h). Furthermore, we found that *Casp8* knockout tumors obtained robust immunotherapy resistance and shortened mouse survival upon immunotherapy compared to EV controls (Fig. 6i, j). These results suggest that targeting cell apoptosis pathways is a promising strategy to overcome immunotherapy resistance, which is prevalent in many cancer types including PDAC.

## Discussion

While predictors of primary resistance to immunotherapy in patients have been studied in detail[34–36], there has been far less investigation of acquired resistance[37]. Our results suggest that distinct mechanisms operate in the two settings: whereas primary resistance is typically associated with a paucity of T cells and an immunosuppressive TME[16], acquired resistance is paradoxically associated with a hyperimmune TME, where resistance to T cell-mediated cytotoxicity occurs through a change in cell state. Our data suggest that epithelial plasticity (EMT) confers resistance by providing PDAC cells with an intrinsic resistance to T cell killing rather than by assembling an immunosuppressive tumor microenvironment. Mechanistically, EMT drives repression of *Irf6*, making the tumor cells less susceptible to the pro-apoptotic effects of T cell-derived TNF-α.

Preclinical and clinical studies by us and others have provided encouraging results from combination therapies that include CD40 agonists[13–15,17,18,38]. These studies, and our results here, indicate that therapy responses are heterogenous, reflecting various resistance programs. Oncogenic signaling, defects in Ag presentation, and immune-suppressive elements of the TME are known to induce

primary resistance, whereas the mechanisms of acquired resistance are poorly understood and likely involve strategies distinct from those used by primary tumors[39]. This concept is reinforced by our finding that Esc tumors with acquired resistance exhibited a paradoxical increase in CD8[+] T cell infiltration, a reduction in gMDSC infiltration, and a decrease in the expression of the co-inhibitory molecules PD-L1 and CD73 – effects that would all be predicted to enhance rather than diminish antitumor immunity. These findings suggest that once a tumor has responded to immunotherapy, it activates mechanisms other than a reconfiguration of the TME to evade further immune attack.

In the present study, we built a model system to understand the stark differences between primary resistance and acquired resistance and discovered that tumor cells exploit quite different strategies to overcome short-term (primary) and long-term (acquired) immune pressure. In PDAC, primary resistance to immunotherapy, rather than acquired resistance, is the major clinical problem. Thus, assessing the clinical relevance of the mechanisms revealed by our animal model – EMT and *Irf6* repression – necessitated the identification of human datasets for which transcriptomic data was available before and after the emergence of immunotherapy resistance. Consequently, we employed results from a study of NSCLC (the only dataset we were able to find that met these criteria). Our analysis revealed an overlapping pattern associated with acquired resistance to ICB in this different tumor type, supporting the notion that EMT-associated repression of *Irf6* is a generalized mechanism of acquired resistance to immunotherapy. Future studies will be needed to determine whether this mechanism applies to larger human cohorts and other tumor models, including those that are not associated with mutations in *KRAS* or *TP53*.

An inverse relationship between EMT and immunotherapy response has been well-documented in mice and humans[8,10–12,40–42]. Our finding that EMT blunts the tumor cell's response to T cell-derived TNF-α is in line with previous reports implicating TNF-mediated killing as a crucial mechanism of tumor elimination, especially in poor neoantigen-expressing tumors[30,43]. Accordingly, genetic ablation or pharmacological inhibition of the TRAF2/cIAP complex, which facilitates TNF-induced cell death, resulted in improved antitumor responses when combined with ICB in preclinical studies[31,33,44]. In the context of these studies, our work suggests that strategies that re-establish sensitivity to death receptor-mediated killing, as reported recently in the setting of tumor cell intrinsic resistance to CAR T cell killing[45], may reverse or prevent the emergence of resistance after an initial response to immunotherapy. As the OVA system represents an experimental model of antigen response and resistance, it will be important in the future to confirm these EMT-related tumor-immune interactions in the setting of endogenous tumor antigens.

We found that single cell clones gave rise to heterogeneous responses to immunotherapy. This finding is consistent with studies of targeted therapies such as BRAF inhibition, where acquired resistance can result from either the outgrowth of rare subclones (i.e. cells carrying with mutations in the drug target) or non-genetic cell state transitions that enable the outgrowth of cells that are resistant on the basis of an altered phenotype[46]. We previously reported that epithelial-mesenchymal transitions occur early in tumor progression[47], and thus it is possible that pre-existing mesenchymal subclones could contribute to acquired resistance; formally distinguishing between these possibilities will require single cell tracing approaches. Nevertheless, we found that co-culturing OVA-expressing tumor cells with antigen-specific T cells for just 2 days resulted in a profound activation of EMT programs in the surviving cells (Supplementary Fig. 8), suggesting that acquired resistance is the product of plasticity rather than the outgrowth of pre-existing mutant subclones. While we believe this is likely due to the preferential escape from cytotoxicity of cells with a more mesenchymal phenotype, we cannot rule out the possibility that T cells may also possess EMT-inducing properties. In addition, we found that

many Esc lines exhibited *Myc* amplifications, raising the possibility that genetic alterations or copy number changes could contribute to acquired immunotherapy resistance, potentially through direct or indirect effects on EMT.

The Irf family of transcription factors has broad activities in immune function that extend beyond their originally described roles in type 1 interferon responses[48]. Unlike other Irf family members, Irf6's known roles are limited to the proliferation and function of epithelial (epidermal) cells[32,49]. In humans, heterozygous mutations in *IRF6* are associated with van der Woude syndrome, a condition associated with facial malformations due to developmental dysmorphogenesis[50]. Our data show that EMT leads to the repression of *Irf6* – either during spontaneous EMT in Esc cells or EMT induced by the expression of *Zeb1* and *Snail*. *Irf6* ablation in parental cells protects them from T cell killing, while *Irf6* restoration leads to greater T cell killing and immunotherapy responsiveness, effects that may be related to a tumor cell's "TNF cytotoxicity threshold"[31]. Further support for this model comes from our finding that the subset of lung cancer patients whose tumors exhibited decreased *IRF6* expression in the setting of immunotherapy resistance also exhibited a strong EMT signature.

While it is unclear how Irf6 loss exerts its protective influence at a molecular level, a recent report in fish suggests that Irf6 can reduce the activity of IFN and NF-κB reporters in transfected 293 T cells[51]. As noted by us and others[52,53], it is possible that Irf6 modulates tumor cell sensitivity to death stimuli including TNF in part by participating in or reversing the EMT state. Future studies will be needed to understand how Irf6 regulates TNF cytotoxicity and to determine whether other pathways besides Irf6 and TNF signaling are dysregulated during EMT to contribute to acquired resistance.

## Methods

### Animals

C57BL/6, C57BL/6-Tg(TcraTcrb)1100Mjb/J (OT-I), and NOD/SCID mice were purchased from the Jackson Laboratory and/or bred at the University of Pennsylvania. Mice were housed under a 12h-12h light-dark cycle, temperature of 18–23 °C, humidity of 36–56%, and pathogen-free conditions. All animal procedures used in this study were performed following the National Institutes of Health guidelines. All mouse procedure protocols used in this study were in accordance with, and with the approval of, the Institutional Animal Care and Use Committee (IACUC) of the University of Pennsylvania (protocols 804643 and 805650).

### Cell lines

The 4662 murine PDAC cell line was derived from a spontaneous pancreatic cancer in a female KPC mouse on the C57BL/6 J background as previously described[54]. 4662 cell lines were tested and authenticated by the Research Animal Diagnostic Laboratory (RADIL) at the University of Missouri using the Infectious Microbe PCR Amplification Test (IMPACT). 4662 early progressor (*n* = 2) and escape cell lines (E1 - 8, C7.e1, and C10.e1) were isolated from non-responders and recurrent tumors reaching ≥3 mm mean tumor diameter beyond day 75, respectively, following inoculation of the 4662 mouse PDAC cell line and treatment with chemoimmunotherapy or immunotherapy alone. Similarly, early progressor and escape cell lines were established in the setting of orthotopic 4662 tumors by ultrasound monitoring. Single cell clones were derived from the 4662 PDAC cell line by limiting dilution. 4662 PDAC cell lines with a full length of OVA and tdTomato as a surrogate have been generated as previously described[54]. PDAC cell lines were cultured in a standard cell culture medium including Dulbecco's Modified Eagle's medium (DMEM) with high glucose, 10% heat-inactivated FBS and Glutamax (GIBCO). These tumor cell lines were used for less than 20 passages and tested negative for mycoplasma contamination (MycoAlert Mycoplasma Detection Kit, Lonza).

YFP labeling of cell lines was done with pCDH-CMV-EF1α-YFP, modified from pCDH-CMV-EF1α-RFP (System Biosciences, CD512B-1). We used 293 T cells (Clontech, 632180) for lentivirus packaging. Detection of cytokines and chemokines in tumor culture supernatants was performed using a Proteome Profiler Mouse Cytokine Array Kit (R&D Systems).

## Tumor implantation and therapy response assessment

$5 \times 10^5$ 4662 PDAC cells were implanted subcutaneously (s.c.) into 6-8-week-old female C57BL/6 mice and mice bearing PDAC were randomly allocated to each group and given therapy or control treatment. Some mice received $10^5$ 4662 PDAC cells in 50 µL DMEM orthotopically into the tail of pancreas on a sterile field under anesthesia. For tumor rechallenge, mice with complete response (CR) were administered control or T- or NK-cell depleting Ab and rested for 50 days, followed by subcutaneous inoculation of $10^6$ 4662 PDAC cells. For tumor growth kinetics, tumors were measured every 3-4 days by calipers and represented as the mean values of perpendicular diameters. For long-term survival studies using a subcutaneous tumor model (Figs. 2, 3, 5, 6), endpoint criteria included tumor volume calculated by $0.5236 \times length \times width^2$ exceeding 500 mm³, severe cachexia, or weakness and inactivity, as per our mouse protocols. For the tumor growth curves in Fig. 1 and Supplementary Figs. 1–3, animals were sacrificed when tumors reached a diameter of 2 cm or animals showed clinical signs of deterioration. Tumors were harvested 18-21 days following implantation or a week after treatment for flow and tissue analyses. Therapy response assessment was defined as follows: Durable CR indicated the absence of palpable tumor at the completion of the experiment; CR with relapse indicated CR or near CR ($\leq 1 \times 1$ mm tumor diameter) followed by progressive tumor growth; partial response denoted tumor regression to $\leq 30\%$ of the maximum tumor diameter followed by progressive tumor growth; and finally, non-responders showed no response to therapy, transient stable disease, or limited response not meeting criteria for partial response.

## In vivo treatment and T/NK-cell depletion

The general treatment schedule was previously described[17]. Briefly, mice with mean tumor diameter 5–7 mm (tumor volume around 100 mm³) were enrolled and treated intraperitoneally (i.p.) with anti−PD-1 (clone RMP1-14, 200 µg/dose) for 7 times and anti−CTLA-4 (clone 9H10, 200 µg/dose) for 3 times every 3 days. Gemcitabine (Hospira) and nab-paclitaxel (Abraxane, Celgene) (120 mg/kg each), purchased from the Hospital of the University of Pennsylvania Pharmacy, were administered i.p. one time on the next day of first ICB treatment. Anti-CD40 agonist (clone FGK45, 100 µg/dose) was co-injected i.p. with second dose of ICB. Control animals were treated with equivalent doses of isotype control antibodies.

Depletion of CD4⁺ T cells, CD8⁺ T cells, and NK cells was achieved by i.p. injections of anti-CD4 (clone GK1.5, 200 µg/dose), anti-CD8 (clone 2.43, 200 µg/dose), and anti-NK1.1 (clone PK136, 200 µg/dose) twice weekly from day 50 to 100, respectively. Control groups received isotype control antibodies. Depletions were confirmed by peripheral blood samples and end-of-study flow cytometry, and the efficiencies were approximately 99%. All antibodies administered to mice were from Bio X Cell.

## Adoptive transfer of tumor-specific CD8⁺ T cells

For adoptive transfer studies, $2 \times 10^5$ OVA$^{tdTomato}$-expressing 4662 parental and escape tumors were implanted s.c. into NOD/SCID mice. On day 14 when mean tumor diameter was 5–6 mm, $1.5 \times 10^6$ OVA-specific CD8⁺ T cells prepared from spleen and lymph nodes of OT-I mice using mouse CD8α microbeads (Miltenyi Biotec, 130-117-044) and further flow cytometry sorting, followed by activation with immobilized 2 µg/ml of anti-CD3 (145-2C11; BioLegend) and 1 µg/ml of anti-CD28 (37.51; BioLegend) overnight, were transferred intravenously (i.v.) into tumor-bearing mice.

## Flow cytometry of murine PDAC

For flow cytometric analyses, s.c. tumors were minced and digested in DMEM supplemented with 2 mg/ml of collagenase type IV (GIBCO, 17104-019) and 0.2 mg/ml of DNase I (Sigma, 10104159001) at 37 °C for 45 minutes and filtered through a 70-µM cell strainer to generate single cell suspensions. Cells were then stained with fluorescence-conjugated antibodies and a live/dead stain (Invitrogen, L34966) at 4 °C for 20 minutes and washed twice with cold PBS plus 5% FBS for sample acquisition. For intracellular caspase staining, cells were further permeabilized with Fix/Perm buffer (eBioscience, 00-5523-00) at 4 °C for 30 minutes and stained with active caspase-3 Ab (C92-605, 1:20; BD Biosciences) in Perm buffer (eBioscience) at 4 °C for 30 minutes. Sample acquisition was performed by LSR II flow cytometry (BD Biosciences) and data were analyzed using FlowJo software (Tree Star). Gating strategies for immune cell populations were previously described[16] and shown in Supplementary Fig. 12. Antibodies used for flow cytometry are as follows: anti-mouse CD335 (NKp46) (29A1.4), CD103 (2E7), H-2K$^b$/H-2D$^b$ (28-8-6), H-2K$^b$ bound to SIINFEKL (25-D1.16), F4/80 (BM8), CD45 (30-F11), I-A/I-E (M5/114.15.2), CD11b (M1/70), Ly6C (HK1.4), Ly6G (1A8), CD11c (N418), CD3ε (145-2C11), CD8α (53-6.7), CD4 (RM4-5), CD25 (PC61), CD62L (MEL-14), CD44 (IM7; BD Biosciences), CD324 (E-cadherin) (DECMA-1), CD73 (TY/11.8), CD274 (PD-L1) (10 F.9G2), CD155 (TX56), Arginase 1 (A1exF5; eBioscience), CD80 (16-10A1), CD86 (GL-1), Ki-67 (16A8), CD69 (H1.2F3), CD107a (LAMP-1, 1 ug/ml) (1D4B), IFN-γ (XMG1.2), TNF-α (MP6-XT22), Granzyme B (GB11, 1 ug/ml), and Perforin (S16009A). Antibodies for flow were all from BioLegend and used at 1:100 dilutions unless otherwise indicated.

## Lentiviral transduction of tumor cells for target gene modulation

The pCDH-EF1-FHC vector, a gift from Richard Wood (Addgene plasmid #64874), pCDH-CMV-EF1α-YFP, and pLVX-IRES-zsGreen (Takara Bio) vectors were used for constitutive overexpression. Full-length mouse *Zeb1* and *Snail* were gifts from Thomas Brabletz, University Erlangen, Germany. Murine *Irf6* gene was amplified based on the cDNA template from parental tumors. Point mutations in *Irf6* gene were performed using Q5 Site-Directed Mutagenesis Kit (NEB, E0552S) with mutagenic primers according to manufacturer's instructions and confirmed by sequencing.

The CRISPR vectors, lentiCRISPR v2 (a gift from Feng Zhang [Addgene plasmid #52961]) and LRG2.1 (a gift from Christopher Vakoc [Addgene plasmid #108098]), were used for target gene deletion. We further replaced GFP into YFP in the LRG2.1 vector. The sgRNA sequences were selected using a CRISPick tool (Broad Institute) and cloned into CRISPR vectors using a *BsmBI* restriction enzyme following the instruction from Addgene. CRISPR sgRNA sequences were listed in Supplementary Table 2.

Cloned plasmids were then co-transfected with pVSV-G (Addgene plasmid #8454) and psPAX2 (Addgene plasmid #12260) lentiviral packaging plasmids into 293 T cells (Clontech) using polyethylenimine (PEI; Polysciences, 23966-100) in a ratio of 4:2:2 for plasmid DNA:pVSV-G:psPax2. Lentiviral particles were collected 72 hours after transfection and passed through a 0.45 µm PVDF filter for usage. Tumor cells were transduced with filtered viral supernatants in the presence of 8 µg/ml polybrene (Sigma, H9268) for 24 h, expanded for a couple of days, and selected with 8 µg/mL puromycin (Invitrogen, A1113803) for 5-7 d. YFP⁺ cell sorting by flow cytometry was further conducted for double-gene modulation. Overexpression and knockout efficiencies were assessed by gene-specific qPCR analysis of target genes or western blotting.

## Quantitative real-time PCR (qPCR)

RNA was isolated from cultured tumor cells using the NucleoSpin RNA Kit (Takara Bio) and reverse-transcribed by the High-capacity cDNA

Reverse Transcription Kit (Life Technologies). Diluted cDNA was used for qPCR, which was performed with SsoAdvanced SYBR master mix (Bio-Rad) and the CFX384 Real-Time System (Bio-Rad). Results were normalized to *Tbp* expression using the Bio-Rad software. Primer sequences used for qPCR were in Supplementary Table 2.

### Tumor and CD8+ T cell co-culture

OVA-specific CD8+ T cells from OT-I mice were sorted by CD8α MACS enrichment, followed by CD3ε+CD8α+ cell sorting using flow cytometry, and maintained or activated overnight by pre-coated anti-CD3 (2 μg/ml) and anti-CD28 (1 μg/ml) in a RPMI 1640 culture medium supplemented with 10% FBS, 2.5% HEPES, 1% sodium pyruvate, 1% non-essential amino acids, 1% penicillin/streptomycin (Corning), and 0.1% 2-mercaptoenthanol (all from GIBCO unless otherwise indicated). Activated OT-I cells were co-cultured with OVA$^{tdTomato}$-transduced tumor cells at indicated ratios with or without 5 μg/ml of TNF-α neutralizing antibody (MP6-XT22; BioLegend), and two days later, T-cell killing was measured using Annexin V Apoptosis Detection Kit (eBioscience, 88-8007-72) and 7-AAD Viability Staining Solution (Bio-Legend, 420404) according to the manufacturer's protocol by flow cytometry. Birinapant (MedChem Express, HY-16591) was added to some co-culture from tumor cell plating 1 d prior to T cell co-culture.

For T-cell priming assay, OVA-specific naïve CD3ε+CD8α+CD44$^{lo}$CD62L$^{hi}$CD25$^−$ T cells were sorted by flow cytometry, labelled with 5 μM CellTrace Violet (CTV; Invitrogen, C34557), and co-cultured with OVA$^{tdTomato}$-expressing tumor cells, treated with or without 100 ng/ml of IFN-γ (Peprotech, 315-05) overnight. Three days later, CTV dilution and CD44 and CD25 expression were analyzed on tdTomato$^−$CD8α$^+$ gated OT-I cells by flow cytometry.

### RNA-seq

RNA was prepared from tumor cells at steady states, 1 d after co-culture, and 2-3 weeks following implantation as described above. RNA-seq libraries were prepared either by Novogene (California, USA) or with the NEBNext Ultra II RNA Library Prep Kit (NEB, E7770S) according to the manufacturer's protocol. Libraries were then sequenced on Illumina next generation sequencers, generating either 150 bp paired end or 100 bp single end reads, with generally 20 ~ 25 million reads per sample. Salmon v1.8.0[55] was used to generate raw counts in transcripts per million (TPM) through quasi-alignment to the mm39 reference genome using standard settings. The raw count matrix was subsequently imported into R-studio (R v4.1.2) for downstream normalization and differential gene expression analysis using DESeq2[56]. The differentially expressed gene sets in *Zeb1/Snail* OE compared to EV tumors were included in Supplementary Data 1. Genes were ranked by their Wald statistic for pre-ranked gene set enrichment analysis (GSEA)[57,58]. Multiple published datasets were analyzed in the same manner. De novo and known motifs were identified within 500 bp of promoters of differentially expressed genes using HOMER's (v4.11) findMotifs.pl command[59].

### ATAC-seq

**Library construction.** Libraries were prepared as previously described with minor modifications[60]. Briefly, concentrated Tn5 transposase (Diagenode) was diluted 10-fold using Tn5 dilution buffer (50 mM Tris HCl pH 7.5, 100 mM NaCl, 0.1 mM EDTA, 1 mM DTT, 0.1% NP-40, and 50% glycerol). Transposomes were assembled by loading the diluted Tn5 with the following Illumina sequencing adapters:

Read1 - TCGTCGGCAGCGTCAGATGTGTATAAGAGACAG
Read2 - /5Phos/GTCTCGTGGGCTCGGAGATGTGTATAAGAGACAG
Reverse - /5Phos/C*T*G*T*C*T*C*T*T*A*T*A*C*A*/3ddC/

Nuclei were isolated from 50,000 cells, followed immediately by transposition at 37 °C for 30 min. Transposed DNA fragments were purified using a Qiagen MinElute Kit, barcoded with primers based on Illumina TruSeq indices, and PCR amplified for 5 cycles using NEBNext

High Fidelity 2x PCR master mix (NEB). Libraries were column-purified with the Qiagen PCR Cleanup kit, followed by 1.0x AMPure bead cleanup. Library quality was assessed on 4200 TapeStation (Agilent), and concentrations were quantified by Qubit D1000 assay (Thermo-Fisher Scientific). Samples were sequenced using 150-cycle High Output NextSeq kits (Illumina, 20024907) to generate 75 bp paired end reads.

**Analysis.** Adapters were trimmed with Cutadapt v3.5 and reads were aligned to the mm39 mouse reference genome with Bowtie2 v2.4.4[61]. Sambamba v0.7.1[62] was used to filter out duplicates, while SAMtools v1.9[63] was used to identify and discard reads that aligned to the mitochondrial genome. Peaks were called with Genrich v0.6.1 (https://github.com/jsh58/Genrich) using standard settings in ATAC-seq mode and blacklisted regions were removed with BEDtools v2.30.0[64]. Overlapping peaks were identified with BEDtools, then merged. Raw read counts were determined with featureCounts v2.0.1[65], then imported into R Studio for normalization and differential analysis using DESeq2. This resulted in the detection of approximately 50 K total peaks and 16K - 37K peaks with differential accessibility in each group. Differentially accessible loci were annotated to genomic features using ChIPseeker[66]. Genes and enriched biological pathways associated with differentially accessible loci were determined by GREAT[67]. Motif discovery within differentially accessible loci was done with HOMER's findMotifsGenome.pl command.

Putative target promoters of transcription factors of interest were determined by overlapping mm39 promoters identified through ChIPseeker with known TF motifs. These target promoters were then intersected with the differentially accessible loci and annotated to their nearest genes using a mm39 annotations file and BEDOPS[68]. Genes were de-duplicated and their raw TPM counts were extracted from prior RNA-seq analysis. Genes were then filtered for fold changes of >=2 or <=−2 and plotted by log2TPM.

For gene track visualization, scaled bigwig files were generated using deepTools v3.5.1[69]. Scaling factors were determined from edgeR's[70] calcNormFactors function using raw counts that were derived from featureCounts v2.0.1. Bigwig tracks were visualized using IGV's[71] genome browser and Gviz[72] in R Studio.

### scRNA-seq analysis

Published raw single cell RNA-seq data derived from patient samples were analyzed using the 10x and Seurat v4 pipelines[73]. Briefly, raw counts were determined with Cell Ranger using the hg19 reference genome, and imported into R Studio for analysis using Seurat, as previously described. Data were initially filtered to include cells with at least 100 genes and all genes in at least 3 cells. Samples were merged and further filtered by mitochondrial read percentage and total transcripts. Samples were then integrated and normalized, and variable genes were determined. Cell subpopulations were clustered based on the expression of certain gene markers. The epithelial cell cluster was further subsetted by the mean expression level of certain genes of interest, and the top differentially expressed genes were identified in the high (>mean) and low (<mean) subsets. These differentially expressed genes were used as gene sets for downstream GSEA.

### Immunofluorescence, immunohistochemistry, and H&E staining

Tissues were fixed in Zinc-formalin and embedded in paraffin for histological analysis and immunofluorescence (IF) staining. For IF staining, sections were deparaffinized, rehydrated, and prepared by antigen retrieval. They were then blocked in PBS with 0.3% Triton-X and 5% donkey serum for 1 hour, stained with primary and secondary antibodies, and mounted with Aqua Polymount (Polysciences). Primary antibodies used include chicken anti-GFP (Abcam, ab13970) and rabbit anti-cleaved caspase-3 (Cell Signaling Technology, 9661). Slides were

visualized using an Olympus IX71 inverted multicolor fluorescent microscope equipped with a DP71 camera. ImageJ FIJI software was used for quantification, with each data point an average of 2-3 fields per tumor section.

For immunohistochemistry (IHC) staining, sections were processed as IF staining and quenched with endogenous peroxidase in methanol for 15 min, and then blocked in PBS with 1% BSA for 1 hour. Slide sections were stained subsequently with rat anti-E-cadherin (Takara Bio, M108), rabbit anti-Vimentin (Cell Signaling Technology, 5741), rabbit anti-Twist1/2 (GeneTex, GTX127310), rat anti-Ly6G (Stem Cell Technologies, 60031BT.1), or rabbit anti-CD8a (Cell Signaling Technology, 98941, 1:100) antibodies, followed by biotinylated corresponding secondary antibodies (Jackson ImmunoResearch) and Vectastain Elite ABC-HRP Kit (Vector Laboratories, PK-6100). Sections were developed with DAB Substrate Peroxidase Kit (Vector Laboratories, SK-4100), counterstained with hematoxylin, dehydrated, and mounted with Aqua Polymount (Polysciences). Primary antibodies were used at 1:200 dilution unless otherwise indicated. For H&E staining, sections were deparaffinized, rehydrated, stained with hematoxylin, differentiated with acidic ethanol, stained for eosin, dehydrated, and mounted with Permount. Both IHC and H&E slides were visualized using an Olympus BX41 microscope equipped with a DP25 camera.

### Chromatin immunoprecipitation (ChIP)-qPCR assay

Confluent tumor cells in 15 cm plates were crosslinked with 1.11% formaldehyde for 10 min at room temperature, quenched with 125 mM glycine, and collected with PIPES nuclei isolation buffer by cell scraper. Cells were spun down and dissolved pellets in ChIP lysis buffer containing 10% SDS, 10 mM EDTA, and 50 mM Tris-HCl (pH 8) in distilled water. Proteinase inhibitors were added at all steps. DNA fragmentation was achieved by Bioruptor (Diagenode), with 15 cycles of high-intensity sonication for 5 min at 30 sec interval on/off mode. Sheared chromatins were obtained from supernatants following high-speed centrifugation, diluted, and pre-cleared using Protein G Dynabeads (Thermo Scientific, 10003D) for 2 ~ 3 hours at 4 °C. Unbound samples were subsequently conjugated with control IgG, rabbit anti-ZEB1 (Proteintech, 21544-1-AP), or rabbit anti-SNAIL (Proteintech, 13099-1-AP) antibodies overnight at 4 °C. 10% of pre-cleared samples were set aside for input. Next day, antibody-conjugated protein-DNA complexes were collected by Protein G Dynabeads, with Dynabeads binding for 3 ~ 4 hours at 4 °C, sequential washing by low salt, high salt, LiCl, and TE buffers, and eluting by 2% SDS and 100 mM NaHCO$_3$ in distilled water. For reversal of crosslinking and removing proteins, eluted DNA-protein complexes were incubated with 200 mM NaCl plus RNase A overnight at 65 °C, followed by additional incubation with 40 mM Tris-HCl (pH 7.5), 10 mM EDTA plus proteinase K for 2 hours at 45 °C. Prepared chromatin DNA was further purified using PCR purification Kit (Qiagen). Input DNA was also obtained by same process. The amounts of ZEB1 and SNAIL binding to *Irf6* gene were determined by qPCR analysis using prepared chromatin DNA and primers for specific *Irf6* gene regions (Supplementary Table 2), and calculated by percent of input DNA.

### Cell proliferation and viability assay

For checking cell proliferation, $10^4$ tumor cells were seeded in each 12-well in triplicates and cell density at indicated time points was measured by staining with Hoechst 33342 Solution (Thermo Scientific, 62249) and detecting by spectrometry. Data were normalized to background control and calculated by percent of cell growth compared to day 0.

Cell viability in response to TNF was determined as previously described[43] with some modification. Briefly, $3–5 \times 10^3$ tumor cells were plated in a 96-well plate and treated with 0.2 μg/ml of IFN-γ (Peprotech, 315-05) plus 1 μg/ml of cycloheximide (Cell Signaling Technology, 2112S) and indicated concentrations of TNF-α (BioLegend, 575204). A

pan-caspase inhibitor, Z-VAD-FMK, was purchased from Selleckchem (S7023). Two days later, cell viability was measured by CellTiter-Glo (Promega, G7571) according to manufacturer's instructions. Data were normalized to each group without TNF treatment and a group with the lowest viability. IC$_{50}$ and nonlinear regression curve fits by log(inhibitor) vs. normalized response test were determined using GraphPad Prism 9 (GraphPad).

### Immunoblot analysis

Tumor cells with or without TNF-α treatment or genetic modification were lysed in RIPA lysis buffer with protease and phosphatase inhibitor cocktail (Thermo Scientific, 78444). Equivalent amounts of protein from whole cell lysates were separated by SDS-PAGE and transferred onto PVDF membranes (Bio-Rad). Membranes were blocked in 5% nonfat milk in PBS plus 0.1% Tween-20 and stained with primary antibodies, followed by probing with horseradish peroxidase-conjugated secondary antibodies (Jackson Immunoresearch). Primary antibodies used include goat anti-IRF6 (Novus Biologicals, NBP1-51911), rabbit anti-TRADD (Cell Signaling Technology, 3694), mouse anti-FADD (1F7; Enzo Life Sciences, ADI-AAM-212-E, 1 ug/ml), rabbit anti-Caspase-8 (D35G2; Cell Signaling Technology, 4790), rabbit anti-IκBα (Cell Signaling Technology, 9242), rabbit anti-phospho-NF-κB p65 (Ser536) (Cell Signaling Technology, 3031), rabbit anti-NF-κB p65 (D14E12; Cell Signaling Technology, 8242), and rabbit anti-GAPDH (14C10; Cell Signaling Technology, 2118). Primary antibodies were used at 1:1000 dilution unless otherwise indicated. ECL solution (Thermo Scientific, 32106) was used as a substrate and band signals were detected using ChemiDoc (Bio-Rad).

### Software, statistics, and reproducibility

PRISM and R were used for data processing, statistical analysis, and data visualization. The R language and environment for graphics (https://www.r-project.org) was used in this study for the bioinformatics analysis of RNA-seq and ATAC-seq data. The R packages used for all analysis described in this manuscript were from the Bioconductor and CRAN. Statistical comparisons between two groups were performed using unpaired two-tailed Student's *t*-test. For comparisons between multiple groups, one-way ANOVA with Tukey's HSD post-test was used. For survival comparison between two groups, log-rank (Mantel-Cox) *P* values of Kaplan-Meier curves were determined using GraphPad Prism 9 (GraphPad). On graphs, bars represent either range or standard error of mean (SEM). For all figures, *P* < 0.05 was considered statistically significant, *indicates *P* < 0.05, **P* < 0.01, and ***P* < 0.0001. No statistical method was used to pre-determine the sample size, which was chosen on the basis of previous experimental experience. Data represent at least two independent experiments unless the representative tissue images were obtained from original primary tumors. No data were excluded from the analyses. For measuring tumor phenotypes in vivo, investigators were not blinded to group information, however, the results were reproducible by two independent researchers in different animal facilities. For in vitro experiments, blinding was not required because all the samples were analyzed in a consistent manner.

### Reporting summary

Further information on research design is available in the Nature Portfolio Reporting Summary linked to this article.

## Data availability

The RNA-seq and ATAC-seq data generated in this study have been deposited in the NCBI Gene Expression Omnibus (GEO) under accession code GSE222225. The publicly available data used in a human lung cancer study are available in the NIH dbGap under accession code phs001464.v1.p1. The remaining data are available within the Article, Supplementary Information or Source Data file. Source data are provided with this paper.

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

## Acknowledgements

We thank members of the Stanger and Vonderheide laboratories, as well as the Penn Pancreatic Cancer Research Center, for technical help and scientific discussions. This work was supported by grants from the NIH CA229803 (R.H.V. and B.Z.S), CA076931 (M.S.D.), CA241084 (M.S.D.), the Abramson Family Cancer Research Institute, the Abramson Cancer Center, and the NIH/Penn Center for Molecular Studies in Digestive and Liver Diseases.

## Author contributions

M.S.D., R.H.V., and B.Z.S. conceived the project. I.K., M.S.D., S.Y., R.H.V., and B.Z.S. designed the study. I.K., M.S.D., S.Y., S.B.K., B.M.K., Q.L., J.H.L., J.L., R.J.N., S.K.T., and M.M. performed experiments, analyzed data, and discussed for this study. S.Y., J. W. T., and T.B. performed bioinformatics analysis. T.K. and K.P. provided experimental and human data resources. I.K., M.S.D., S.Y., R.H.V., and B.Z.S. wrote and revised the manuscript. B.Z.S. and R.H.V. supervised this study. All authors discussed the findings and provided input on the manuscript.

## Competing interests

Dr. Stanger receives research funding from Boehringer-Ingelheim and Revolution Medicines and previously served as a consultant to iTeos Therapeutics. Dr. Vonderheide has received consulting fees from BMS; research funding from Revolution Medicines; is an inventor on patients relating to cancer cellular immunotherapy, cancer vaccines, and KRAS immune epitopes; and receives royalties from Children's Hospital Boston for a licensed research-only monoclonal antibody. The remaining authors declare no competing interests.

## Additional information

---

[1]Department of Medicine, Perelman School of Medicine, University of Pennsylvania, Philadelphia, PA, USA. [2]Abramson Cancer Center and Abramson Family Cancer Research Institute, Perelman School of Medicine, University of Pennsylvania, Philadelphia, PA, USA. [3]Department of Cell and Developmental Biology, Perelman School of Medicine, University of Pennsylvania, Philadelphia, PA, USA. [4]Department of Cancer Biology, Perelman School of Medicine, University of Pennsylvania, Philadelphia, PA, USA. [5]Epigenetics Institute, Perelman School of Medicine, University of Pennsylvania, Philadelphia, PA, USA. [6]Penn Genomic Analysis Core, University of Pennsylvania, Philadelphia, PA, USA. [7]Department of Biomedical Sciences, School of Veterinary Medicine, University of Pennsylvania, Philadelphia, PA, USA. [8]Yale Cancer Center, Yale School of Medicine, New Haven, CT, USA. [9]Department of Pathology, Yale School of Medicine, New Haven, CT, USA. [10]Section of Medical Oncology, Department of Internal Medicine, Yale School of Medicine, New Haven, CT, USA. [11]Parker Institute for Cancer Immunotherapy, University of Pennsylvania, Philadelphia, PA, USA. [12]These authors contributed equally: Il-Kyu Kim, Mark S. Diamond, Salina Yuan.
✉e-mail: rhv@upenn.edu; bstanger@upenn.edu

