## [Peer Review File · Nature Communications]

Plasticity-induced repression of Irf6 underlies acquired resistance to cancer immunotherapy in pancreatic ductal adenocarcinomaREVIEWER COMMENTS

Reviewer #1 (Remarks to the Author): with expertise in PDAC, cancer immunology/immunotherapy

The submitted manuscript is on a very important topic. Plasticity is known to play an important role in PDAC development. How plasticity of PDAC contributes to the resistance of PDAC to immunotherapy is not well studied. This manuscript provides strong evidence at the mouse PDAC level by showing that EMT was associated with epigenetic and transcriptional silencing of interferon regulatory factor 6 (Irf6), which renders tumor cells less sensitive to the pro42 apoptotic effects of TNF- α and thus acquire resistance to immunotherapy. However, there are a few critical areas to be addressed.

1) In general, PDAC is considered to have primary resistance to immune checkpoint inhibitor(ICI)-based immunotherapy. Clinically, we rarely see patients except some MSI-high patients who respond to ICI. The clinical impact of this study on the acquired resistance remains to be determined. The author group has led multiple human nivolumab+/-CD40 agonist studies for PDAC. Although chemotherapy was part of study treatments, selected specimen analysis of patients who have durable response vs. non-durable response vs. no response would provide evidence at the human patient level. Plasticity plays a role in the resistance to chemotherapy. The clinical trial specimens would allow assessing the role of the IRF6 pathway in mediating the resistance to both chemotherapy and immunotherapy.

2) Similarly, although this study is focused on the resistance mechanism to immunotherapy, as the clinical impact of acquired resistance to immunotherapy in PDAC is questionable, it would be interesting to study whether the IRF6 mediated resistance mechanism is applicable to the combination of chemotherapy and immunotherapy in the mouse model of PDAC.

3) For the human acquired resistance data, the manuscript analyzed the human NSCLC specimens. However, human NSCLCs are very different from PDAC in their response to ICI.

4) This study shows that myeloid cells are not involved in mediating the role of

plasticity/EMT in the acquired resistance of PDAC to immunotherapy. Myeloid cells may be involved in mediating the role of plasticity in the primary resistance of PDAC to immunotherapy. The mechanism for primary resistance to immunotherapy would be more clinically relevant.

Reviewer #2 (Remarks to the Author): with expertise in PDAC, cancer immunology/immunotherapy

In their manuscript, Kim et al. investigate the molecular and cellular mechanisms that drive the emergence of acquired resistance to immunotherapy in pancreatic ductal adenocarcinoma (PDAC). Using an innovative mouse approach to model and study this phenomenon rarely observed in human PDAC, they discover epithelial-to-mesenchymal (EMT) plasticity as a mechanism of acquired resistance to combinatorial immune checkpoint blockade and CD40 agonism. Mechanistically, they implicate suppression of IRF6 and loss of sensitivity to T cell-mediated and TNF-induced cell death as drivers of resistance in mesenchymal escaper cell lines. While these findings do offer novel insights into tumor intrinsic mechanisms of acquired immunotherapy resistance, it is unclear how dominant and frequent this mechanism of resistance is. Moreover, as nearly all experiments were performed in vitro or subcutaneously in mice, whether this mechanism holds true in the context of an intact immune suppressed PDAC tumor microenvironment is uncertain, casting doubt on the physiological relevance of their findings. Though the study at present is underdeveloped, we believe that addressing the major points below will help to substantiate the authors conclusions and significance of their findings.

Major Comments:

1. As experiments were performed almost exclusively in vitro or in subcutaneous transplantation models, a major limitation of the study is that the mechanisms of resistance were not studied in the context of a heterogenous PDAC tumor arising in its native microenvironment. Though this would ideally be performed in an autochthonous model (e.g. KPC GEMM), orthotopic transplant models would also allow for rapid generation of large cohorts to study treatment response and resistance in the PDAC TME. As such, the

authors should treat either GEMMs or mice bearing orthotopic PDAC tumors with immunotherapy (FCP), and following relapse generate escaper cell lines and determine changes in EMT plasticity, IRF6 expression and signaling, T cell killing, and TNF sensitivity as they have done with subcutaneously derived models. These experiments will confirm that these EMT-related mechanisms of resistance hold true in immune suppressed tumors that arise in the pancreas TME.

2. In Figure 1 the authors treat mice with a combination of chemo- (GA) and immunotherapy (FCP) and derive cell lines from escapers to study the mechanisms of relapse. As it has been shown by others that EMT is a mechanism of chemotherapy resistance in PDAC (Zheng et al. Nature 2015), it is possible that EMT arises as a result of chemotherapy rather than immunotherapy treatment, which perhaps could explain why FCP seems to outperform GAFCP in number of complete responders (ED Fig. 1). The authors should repeat their initial studies in Fig. 1 and generate escaper lines following just FCP treatment to determine whether induction of EMT is an acquired resistance mechanism to solely immunotherapy treatment.

3. In general, the exact treatments and treatment frequencies used in vivo need to be made clear in text and figures. (A) Were mice treated continuously with chemotherapy and immunotherapy regimens? At which timepoints did they receive therapy? This should be marked in the figures. (B) Which combination therapy was used in Fig. 1e-g? (C) What type of ICB did lung cancer patients in Fig. 5e receive? (D) What immunotherapy was given in ED Figs. 5b,c?

4. The mechanisms of resistance were studied in escaper cell lines derived from resistant tumors rather than in the original tumor tissue itself. The authors should stain primary tumor tissues harvested from escaper tumors for EMT markers and IRF6 expression, as well as immune markers for different myeloid and lymphocyte populations to confirm the same resistance mechanisms in situ. This should be performed on the 8 escaper samples from which they derived cell lines to also determine the frequency of these EMT and IRF6-driven mechanisms of resistance compared to other mechanisms.

5. Though the authors generated 4 Ctrl, 8 ESC, and 2 EP lines following treatment, it is unclear how many of these lines were used in subsequent in vitro and transplantation experiments. For example, in Fig.1e, is only 1 cell line each for Ctrl, ESC, and EP used, or multiple different cell lines? Importantly, if there were 8 ESC lines generated, did all of them show the same upregulation of EMT markers and phenotypes, suppression of IRF6, and resistance to T cell killing and TNF-induced cell death? How consistent and frequent is this mechanism of resistance? In addition, expression of EMT markers and IRF6, and resistance to T cell killing and TNF-induced cell death should also be interrogated in EP lines as well.

6. While experiments were performed in ED Fig. 1e-f to try to rule out the possibility that pre-existing resistant clones may give rise to acquired resistance, since the authors did not use an autochthonous model where heterogeneous epithelial and EMT tumor cell populations spontaneously arise, or transplant KPC cell lines with intermixed epithelial and EMT populations, they cannot fully rule out the possibility that existing clones can give rise to de novo resistance to immunotherapy, and should note this in the text and discussion.

7. Do escaper tumors acquire additional genetic alterations or copy number changes that give rise to immunotherapy resistance? The authors should perform whole exome sequencing and copy number analysis of tissues or cell lines from escapers to rule out the possibility that acquisition of new genetic alterations could also give rise to resistance.

8. The analysis of T cell priming and activation in the setting of escaper tumors or tumors expressing EMT TFs is insufficient. First, in Fig. 3a-b and ED Fig. 3a-b, total T cell numbers are shown, but T cell activation markers (Ki67, CD69, CD44) and effector functions (GZMB, CD107a, TNFa, IFNy) are not accessed. It will be critical to know if T cells are activated in these settings. Moreover, as the authors show that escaper and Zeb1/Snail OE lines are resistant to TNF-induced cell death, it will be important to know if T cells (as well as NK cells) also produce less TNF in these resistance settings. It is possible that resistance could result from a combination of both less T cell activation and secretion of TNF as well as tumor intrinsic TNF insensitivity, which the authors will need to interrogate. Immunohistochemical analyzes of T cell numbers, activation markers, and TNF production will also be informative in primary tumor sections from which these escaper lines were derived. As FCP treatment

should increase T cell priming, T cell activation markers and TNF production should be assessed following FCP immunotherapy in mice with parental and escaper tumors.

9. Though experiments in Fig. 2 and 3 clearly demonstrate that EMT plasticity leads to resistance to immunotherapy and T cell killing, data in Fig. 5c-d show that IRF6 overexpression only has a marginal effect on increasing FCP sensitivity in escaper lines, with only 2/10 mice having CRs. Moreover, only 3/7 lung cancer patients that had acquired resistance to immunotherapy had a decrease in IRF6 expression, suggesting in patients other resistance mechanisms may be more predominant. This questions how important and significant IRF6 loss is to immunotherapy resistance mediated by EMT. Perhaps performing in vivo experiments using parental lines with IRF6 knockout may show a more profound effect of the role of IRF6 loss in immunotherapy resistance and are worth pursuing. Given that IRF6 OE seems to increase epithelial markers and decrease mesenchymal markers (ED Fig.7c), and its documented role in regulating E-cadherin (Antiguas et al J Invest Dermatol 2022) and Zeb1 (Li et al Front Oncol 2019), the authors should acknowledge that the impact of IRF6 on immunotherapy outcomes and T cell sensitivity may be in part due to its regulation of the EMT state.

10. Data in Fig. 6 nicely demonstrates that *Irf6* re-expression sensitizes escaper cells to TNF-induced cell death and T cell killing. Since IRF6 re-expression alone has minimal effects on tumor responses or survival following immunotherapy in vivo (Fig. 5c,d), could activating TNF-induced cell death through treatment with cIAP inhibitors or other small molecules be a more potent and clinically relevant approach to overcome immunotherapy resistance? Similarly, does TNF α blockade in mice with transplanted parental KPC cells lead to resistance to FCP? These experiments would further demonstrate the importance of sensitivity to TNF-induced cell death in EMT-mediated acquired resistance and test a strategy to overcome immunotherapy resistance in PDAC.

Minor Comments:

1. Which cell line was transplanted in Fig. 3d? This is not stated.

2. TPM needs to be defined.

3. Need to show IF staining images for Fig. 6c

Reviewer #3 (Remarks to the Author): with expertise in PDAC, immunology, epigenetics, EMT

In this study, Kim and colleagues analyzed the acquired resistance to immunotherapy. Briefly, the authors generated acquired resistance models for GAFCP, reporting that EMT-transcription factors Zeb1 and Snail modulated the tumor cell intrinsic resistance to T cell killing by Irf6. Overall, this is an interesting concept, and the findings are potentially important and helpful to advance our current knowledge of pancreatic cancer treatment. Overall, the quality of the work is high and most of the conclusions are supported by the results. However, the concept/phenomenon that epithelial-mesenchymal plasticity determines immune escape and resistance to immunotherapy in cancer has been reported in several studies and reviews, even though the authors emphasize the acquired resistance. In terms of writing, some results are lack of detailed description, which make it really hard to follow. And some descriptions are overstated.

Major points:

1. P5 line 84 "Antitumor responses were associated with immunological memory that was dependent upon T cells but not NK cells (Extended Data Fig. 1b–d). Thus, the combination of CD40 agonist and ICB leads to durable CRs and T cell memory in PDAC". Flow cytometric analysis of T cells should be included to support this conclusion. The treatment timeline should be included in the legend. This also applies to other similar figures.

2. Since only one cell line 4662 was used in this study, does the "plasticity" rely on mutations of KRAS and TP53 status? Should be discussed.

3. For NK and CD4/8 depletion assays, the efficiency should be included.

4. In Figure 2, considering the existence of many EMT-TFs, why did the authors choose Zeb1

and Snail for the follow-up study? The downregulated or upregulated gene sets used for GSEA analysis should be provided as supplementary data (Figure 2d, e)

5. The proportion of tumor-infiltrating gMDSCs, DCs and CD8 T cells were remarkably changed in Esc, and by Zeb1/Snail OE or KO. How about the phenotype of myeloid cells and the cytotoxicity of T/NK cells? Moreover, the authors also need to analyze whether and how Esc and Zeb1/Snail OE/KO cells directly influence the trafficking or function/activity of those altered immune cells.

6. The analysis detail and result of sequencing data (ATAC, RNA and single cell) should be obtained, in which some important part should be explained in the main text, such as sequencing depth, numbers of detected peaks, and/or information of peaks with differential accessibility. More seriously, the quality of ATAC-seq may be challenged, for too few peaks as common sense. In addition, is there any EMT-related genes changed in the ATAC-seq and/or RNA-seq?

7. In figure 4, more information about Irf6 signature genes should be provided.

8. Mechanistically, how does Zeb1/Snail OE regulate Irf6 at the epigenetic and transcriptional level? For example, whether Zeb1/Snail1 directly bind to Irf6-encoding sequence and inhibit its transcription; The large change in chromatin accessibility during EMT plasticity could be mediated by EMT-TFs? Or whether the transcriptional repression and accessibility-loss were driven by histone modification or DNA methylation.

9. In Figure 5, author used an immunotherapy cohort from lung cancer patients instead of PDAC patients, please provide rationale or at least describe the reason in the manuscript.

10. Interestingly, agonistic anti-CD40 Ab could bring about a certain percentage of tumor regressions and responses. Could the authors explain or at least hypothesize the relevance between Esc/EMT/Zeb1/Snail1/Irf6 and CD40 axis?

Minor concerns

1. Figure 1a, the color line should be removed based on my understanding.
2. Figure 1d, "Mice with recurrent tumor after CR.....", here which group(treatment) it means, or author combine all treatments?
3. P5 line 82 "However, combination regimens that included agonistic anti-CD40 Ab (F) resulted in tumor regressions and prolonged survival, including complete responses (CRs) "the description of result is not accurate.
4. To many experimental data were showed only by box graph or dot plot (i.e. Fig 3a-b, 3g-j, S3a-b, 6b-d). The representative flow graph and histopathological figures would improve the credibility.
5. I highly suggest the authors pay more attention on the English writing to avoid the overstatement. There are some unclear descriptions in Result section, Figure legend, and especially Method, such as Fig 5f-g. Logic and grammar also need to be improved.

Reviewer #4 (Remarks to the Author): with expertise in PDAC, cancer immunology/immunotherapy

This manuscript describes a novel mechanism of immunotherapy resistance whereby EMT-associated transcription factors silence IRF6, resulting in tumor intrinsic insensitivity to TNF-alpha and T cell mediated killing. They use as a model pre-clinical therapy with agonistic CD40 Ab combinations, which are relevant to the disease. They also employ a series of PDAC models involving cell lines derived from mice differential response to immunotherapy. They later deduced from GSEA that Zeb1 and Snail overexpression were linked with this escape from immunotherapy efficacy. A variety of mechanistic studies were also performed whereby OT-1 T cells were used to define a T cell defect that was indeed tumor intrinsic. There is also some data from lung cancer patients establishing EMT signatures are enriched in those who do not benefit from immunotherapy to further support their hypothesis. Overall, the manuscript is well written and adds to the field. The data included are quite rigorous with evidence of convincing group sizes and reproducibility. There are a few suggestions below:

1. It is unclear why the data from orthotopic PDAC models is in the supplement as opposed to the main body of the manuscript. This is a better model and would likely be more of interest to the field if it is more prominently featured in the manuscript. The authors should consider moving it.

2. There is no mention of the increase of CD4+ T cells in Figure 3b of the results. This did not seem to happen in the orthotopic model (in supplement). Was it an artifact, or how do the authors explain this result?

3. In Figure Legend 1, line 870 is it a typo and should there be an "A" in the "GFCP" abbreviation? Or was abraxane not given?

4. Discussion, the manuscript is focused primarily on TNF- α mediated killing but other mechanisms of T cell mediated cell death may be operative. Can the authors comment on the role of Perforin and/or granzyme as alternatives and if they suspect IRF6 may influence these other modes of cell killing?

5. There are inherent limitations of using an OVA-based system to study antigen specific responses in the context of tumors. These should be more appropriately acknowledged as a limitation in the discussion, since the results were not validated with a true endogenously arising tumor antigen. Likewise the generalizability of the results from lung cancer patients to patients with pancreatic cancer is not discussed. These two tumor types are quite different and it is understandable how the data fits in this paper, however this is not acknowledged.

Point-by-point Response to the Reviewers

RE: MS# NCOMMS-23-23245-T

"Plasticity-induced repression of Irf6 underlies acquired resistance to cancer immunotherapy"

Reviewer #1 (Remarks to the Author): with expertise in PDAC, cancer immunology/immunotherapy

The submitted manuscript is on a very important topic. Plasticity is known to play an important role in PDAC development. How plasticity of PDAC contributes to the resistance of PDAC to immunotherapy is not well studied. This manuscript provides strong evidence at the mouse PDAC level by showing that EMT was associated with epigenetic and transcriptional silencing of interferon regulatory factor 6 (Irf6), which renders tumor cells less sensitive to the pro-apoptotic effects of TNF- α and thus acquire resistance to immunotherapy. However, there are a few critical areas to be addressed.

Response: We appreciate the positive feedback.

1) In general, PDAC is considered to have primary resistance to immune checkpoint inhibitor(ICI)-based immunotherapy. Clinically, we rarely see patients except some MSI-high patients who respond to ICI. The clinical impact of this study on the acquired resistance remains to be determined. The author group has led multiple human nivolumab+/-CD40 agonist studies for PDAC. Although chemotherapy was part of study treatments, selected specimen analysis of patients who have durable response vs. non-durable response vs. no response would provide evidence at the human patient level. Plasticity plays a role in the resistance to chemotherapy. The clinical trial specimens would allow assessing the role of the IRF6 pathway in mediating the resistance to both chemotherapy and immunotherapy.

Response: We appreciate these constructive comments and fully agree that primary resistance rather than acquired resistance to immunotherapy is the major clinical issue in PDAC. Thus, although we used PDAC cell lines, these reagents are best viewed as a model system to understand the stark differences between primary resistance and acquired resistance in cancer more generally rather than addressing the clinical dilemma in PDAC specifically. We have addressed this point in the revised Discussion.

As the reviewer correctly points out, our group has previously studied the clinical response of patients to CD40 agonist-based therapy (O'Hara et al., 2021; Padron et al., 2022). In that work, distinct subsets of patients showed objective responses to chemoimmunotherapy regimens, and we suggested potential biomarkers for the responsiveness to each chemoimmunotherapy by retrospective analyses. Post-treatment biopsies on PRINCE trials were all cycle 2 on-study; hence, no potentially informative samples (i.e. following response and relapse) were obtained. Meanwhile, in tumor transcriptome datasets from 12 pancreatic cancer patients (who received gemcitabine/nab-paclitaxel plus nivolumab and/or CD40 agonist thereafter), we observed positive and negative correlations between CDH1/ZEB1/SNAIL and IRF6

that are consistent with what we observed in our mouse study, as well as evidence for the connection between IRF6 and CASP3/8 (**Response Figure 1a-e**). Given that this study involved patients receiving therapy up front and thus assessed only primary resistance, the failure to observe a correlation between expression of CDH1, ZEB1, SNAIL, and IRF6 on overall survival (OS) (**Response Figure 1f-i**) is not surprising. Similarly, we could not disentangle the effects of EMT/IRF6 on chemotherapy vs. immunotherapy responses since all patient groups received combined chemoimmunotherapy.

Response Figure 1. Transcriptome analysis of clinical PDAC specimens and association with OS status. Correlations between IRF6 vs. CDH1 (a), ZEB1 (b), SNAIL (c), CASP3 (d), or CASP8 (e) in the transcriptomes of pancreatic cancer. f–i, Expression of CDH1, ZEB1, SNAIL, and IRF6 in patients grouped by OS status. TPM, transcripts per million.

Nevertheless, we have attempted to address the Reviewer’s question in our mouse model by treating tumors with immunotherapy alone and then newly generating additional Esc cell lines. Importantly, we found that these lines also represented EMT phenotypes, downregulation of *Irf6*, and poor responses to immunotherapy (**Response Figure 2a-d**), suggesting that the EMT/IRF6 pathway is an acquired resistance mechanism to immunotherapy in the absence of chemotherapy. We have included **Response Fig. 2a-c** in **Supplementary Fig. 2d-f** of the revised manuscript.

Response Figure 2. EMT/IRF6 downregulation mediates acquired resistance mechanism to immunotherapy in additional pancreatic cell lines. a, Representative bright field (top) and H&E (bottom) images of *in vitro* and *in vivo* Esc lines, generated under CD40 agonist plus anti-CTLA-4 (FC) or anti-CTLA-4 plus anti-PD-1 (FCP) treatment. Scale bars, 250 μ m. Expression of EMT-related genes (**b**) and *Irf6* (**d**) in 4662 parental, EP, and Esc lines generated under chemoimmunotherapy (GAFCP) or immunotherapy alone (FC,FCP). Each dot in EP and Esc lines represents averaged results from each cell line. **c,** Growth of 4662 Esc tumors generated under immunotherapy alone (FC), treated with control IgG or immunotherapy (FCP) ($n = 5$ or 8).

2) Similarly, although this study is focused on the resistance mechanism to immunotherapy, as the clinical impact of acquired resistance to immunotherapy in PDAC is questionable, it would be interesting to study whether the IRF6 mediated resistance mechanism is applicable to the combination of chemotherapy and immunotherapy in the mouse model of PDAC.

Response: As suggested, we performed experiments testing the effect of re-expressing IRF6 on the response of resistant Esc tumors to combined chemoimmunotherapy. In mice with Esc tumors, combination therapy achieved stable disease control in some animals, but no complete responses (CR) were observed. By contrast, expression of IRF6 in Esc cells markedly improved responses, as 4 out of 10 mice with IRF6-expressing Esc tumors achieved durable CRs, with a significant improvement in survival (**Response Figure 3a,b**). These data are included in the revised manuscript (**Supplementary Fig. 10d,e**).

Response Figure 3. *Irf6* restoration improves the response to combination chemoimmunotherapy. Tumor growth (a) and survival (b) of mice bearing 4662 Esc EV and Esc *Irf6* tumors treated with control IgG or combination chemoimmunotherapy (GAFCP) ($n = 10$).

3) For the human acquired resistance data, the manuscript analyzed the human NSCLC specimens. However, human NSCLCs are very different from PDAC in their response to ICI.

Response: We agree and recognize that NSCLCs as a group have a far more favorable response to immunotherapy. As noted above, a major objective of our work was to understand the mechanisms of acquired resistance (a major clinical problem in NSCLC and many other cancers) and how they may differ from primary resistance. Consequently, any human correlative evaluation of the mechanisms our studies revealed – cell plasticity and *Irf6* expression – necessitated investigation of the corresponding clinical scenario. We took great efforts to identify any cohorts in which patients with acquired resistance to ICI had accompanying pre- and post-immunotherapy transcriptome data. The NSCLC dataset from Politi and colleagues was the only study in human cancer we were able to find that met these criteria. We view the fact that we observed an overlapping pattern associated with acquired resistance to ICI in this different tumor type, where one could imagine completely orthogonal resistance mechanisms, as supporting rather than weakening our model. We have included our rationale in the revision.

4) This study shows that myeloid cells are not involved in mediating the role of plasticity/EMT in the acquired resistance of PDAC to immunotherapy. Myeloid cells may be involved in mediating the role of plasticity in the primary resistance of PDAC to immunotherapy. The mechanism for primary resistance to immunotherapy would be more clinically relevant.

Response: We have no doubt that myeloid cells have an important role in primary resistance to immunotherapy, as we have shown in several prior publications using similar platforms (Bayne et al., 2012; Beatty et al., 2015; Li et al., 2018; Markosyan et al., 2019; Winograd et al., 2015). Therefore, it was extremely interesting to us to discover that the immune microenvironment in the setting of acquired resistance had *fewer* immunosuppressive myeloid cells and *more* T cells, a result which further underscores our conclusion that acquired resistance involves a tumor cell-intrinsic mechanism, distinct from the features of the TME that contribute to primary resistance.

For our revision, we sought to further understand the molecular basis of these observations. We found that Esc and *Zeb1/Snail* OE tumor cells express lower levels of myeloid cell-recruiting cytokines and chemokines – including G-CSF, GM-CSF, IL-1a, and CXCL2 – than parental and EV tumor cells, respectively (**Response Figure 4a-c**). These data have been included in **Supplementary Fig. 6a-c** in the revision.

Response Figure 4. Esc and *Zeb1/Snail* OE tumor cells express lower levels of gMDSC recruiting cytokines and chemokines. **a,b**, Levels of the indicated factors measured by cytokine/chemokine array in tissue culture supernatants from 4662 EV and *Zeb1/Snail* OE (ZS) tumor cells. **c**, mRNA levels (TPM) of myeloid cell-recruiting cytokines and chemokines in 4662 parental vs. Esc (top) and 4662 parental EV vs. *Zeb1/Snail* OE (bottom) tumor cells.

By contrast, we observed that both Esc and *Zeb1/Snail* OE tumors exhibited increased macrophage infiltration. The frequency of Arginase I⁺ macrophages was elevated in these resistant mesenchymal tumors compared to parental and EV tumors; however, most tumor-infiltrating macrophages were MHC II⁺ M1 macrophages with high expression of CD80 and CD86 (**Response Figure 5a,b**). Consequently, CD8 T cells in Esc and *Zeb1/Snail* OE tumors showed evidence of intact or even greater activation and cytolytic cytokine production than those in parental and EV tumors (**Response Figure 5c,d**). We have included these data in **Supplementary Fig. 6d-g** of the revised manuscript.

As noted earlier, we agree that the main challenge in PDAC immunotherapy is primary resistance. Indeed, we and others have previously studied primary resistance in detail (citations above) and conclude that the underlying mechanisms are dominated by the

activity of immunosuppressive myeloid cells. The platform used in this study was not intended to reproduce the clinical scenario (since up-front responses are required for acquired resistance to emerge). Rather, the system allowed us to investigate a phenomenon that is pervasive across tumor types, highlighting the conclusion that tumors may use markedly different strategies to overcome short-term (primary) vs. long-term (acquired) immune pressure, with the tumor microenvironment dominating the former and tumor cell intrinsic factors dominating the latter.

Response Figure 5. *Esc* and *Zeb1/Snail* OE tumors exhibit an increased abundance of activated macrophages and cytotoxic T cells. a,b, The frequencies of total, Arginase I⁺, and MHC II⁺ macrophages and MFI of CD80 and CD86 in macrophages in 4662 parental vs. *Esc* (a) and 4662 parental EV vs. *Zeb1/Snail* OE tumors (b). **c,d,** The frequencies of CD8 T cells expressing activation markers and effector molecules among total CD8 T cells in 4662 parental vs. *Esc* (c) and 4662 parental EV vs. *Zeb1/Snail* OE tumors (d) a week post immunotherapy are depicted. Each dot represents a biological replicate.

Reviewer #2 (Remarks to the Author): with expertise in PDAC, cancer immunology/immunotherapy

In their manuscript, Kim et al. investigate the molecular and cellular mechanisms that drive the emergence of acquired resistance to immunotherapy in pancreatic ductal adenocarcinoma (PDAC). Using an innovative mouse approach to model and study this phenomenon rarely observed in human PDAC, they discover epithelial-to-mesenchymal (EMT) plasticity as a mechanism of acquired resistance to combinatorial immune checkpoint blockade and CD40 agonism. Mechanistically, they implicate suppression of IRF6 and loss of sensitivity to T cell-mediated and TNF-induced cell death as drivers of resistance in mesenchymal escaper cell lines. While these findings do offer novel insights into tumor intrinsic mechanisms of acquired immunotherapy resistance, it is unclear how dominant and frequent this mechanism of resistance is. Moreover, as nearly all experiments were performed in vitro or subcutaneously in mice, whether this mechanism holds true in the context of an intact immune suppressed PDAC tumor microenvironment is uncertain, casting doubt on the physiological relevance of their findings. Though the study at present is underdeveloped, we believe that addressing the major points below will help to substantiate the authors conclusions and significance of their findings.

Response: We thank Reviewer #2 for the positive comments.

Major Comments:

1. As experiments were performed almost exclusively in vitro or in subcutaneous transplantation models, a major limitation of the study is that the mechanisms of resistance were not studied in the context of a heterogenous PDAC tumor arising in its native microenvironment. Though this would ideally be performed in an autochthonous model (e.g. KPC GEMM), orthotopic transplant models would also allow for rapid generation of large cohorts to study treatment response and resistance in the PDAC TME. As such, the authors should treat either GEMMs or mice bearing orthotopic PDAC tumors with immunotherapy (FCP), and following relapse generate escaper cell lines and determine changes in EMT plasticity, IRF6 expression and signaling, T cell killing, and TNF sensitivity as they have done with subcutaneously derived models. These experiments will confirm that these EMT-related mechanisms of resistance hold true in immune suppressed tumors that arise in the pancreas TME.

Response: We appreciate these constructive comments and agree that a limitation of our study as presented was its reliance on subcutaneous models. Thus, as suggested, we created additional cohorts of mice bearing orthotopic 4662 parental tumors and treated them with immunotherapy (FCP starting from day 8), as was done for our subcutaneous tumor model. Some mice received control IgG (to establish control cell lines). We monitored tumor growth by ultrasound, identified mice with complete responses (CRs), and observed them for recurrence. Using this strategy, we were able to generate 4 Esc cell lines from relapsed tumors in the immune-suppressed orthotopic microenvironment (~2-3 months after treatment); consistent with our earlier findings, these newly-derived lines all exhibited a mesenchymal morphology, downregulation of

E-cad and *Irf6*, and upregulation of EMT-related genes (**Response Figure 6a,b**). Moreover, these additional Esc tumors exhibited poorly differentiated histology, in contrast to the well-differentiated histology of control tumors (**Response Figure 6a**). In summary, tumors that relapse following successful immunotherapy in the orthotopic setting exhibit the same changes in cell state (i.e., EMT) as tumors that acquire resistance in the subcutaneous setting.

Response Figure 6. EMT drives immunotherapy resistance in the pancreatic environment. **a**, Representative bright field (top) and H&E (bottom) images of *in vitro* and *in vivo* (orthotopic) control and Esc tumors following treatment with control IgG and immunotherapy (FCP), respectively. Scale bars, 250 μ m. **b**, qPCR of E-cad, EMT-related genes, and *Irf6* in control and Esc tumor cell lines established from orthotopic tumors. **c-e**, Mean tumor diameter over time by ultrasound imaging of orthotopic 4662 EV or *Zeb1/Snail* OE tumors treated with control IgG or immunotherapy (FCP) (**c**). Bright field dissection images (**d**) and tumor weights (**e**) of orthotopic pancreas tumors 6 weeks post tumor implantation ($n = 9-10$ /group).

We agree with the Reviewer that the pancreatic microenvironment could be more fully explored as a potential variable in the resistance phenotype. To this end, we secondly implanted subcutaneous tumors (EV or *Zeb1/Snail* OE isogenic pairs) into the pancreas to determine whether similar patterns of sensitivity/resistance emerged in the more immunosuppressive orthotopic microenvironment. Similar to our findings with subcutaneous tumors, EV tumors regressed upon immunotherapy (FCP starting day 10), whereas *Zeb1/Snail* OE tumors progressively grew upon immunotherapy (**Response Figure 6c-e**). Although we did not reproduce all the findings in this model (it will take additional months to more than a year), we believe these experiments add further confidence that our observations hold true across different tumor microenvironments. These data are included in **Figure 2h-I** of the revision.

2. In Figure 1 the authors treat mice with a combination of chemo- (GA) and immunotherapy (FCP) and derive cell lines from escapers to study the mechanisms of relapse. As it has been shown by others that EMT is a mechanism of chemotherapy resistance in PDAC (Zheng et al. Nature 2015), it is possible that EMT arises as a result of chemotherapy rather than immunotherapy treatment, which perhaps could explain why FCP seems to outperform GAFCP in number of complete responders (ED Fig. 1). The authors should repeat their initial studies in Fig. 1 and generate escaper lines following just FCP treatment to determine whether induction of EMT is an acquired resistance mechanism to solely immunotherapy treatment.

Response: We appreciate these points. We administered only a single dose of chemotherapy (whereas immunotherapy treatment was more prolonged) and obtained escape variants under the GAFCP regimen almost 60 days following chemo treatment. Therefore, we believe it is unlikely that late escape variants represent an escape from chemotherapy. More importantly, we have confirmed that escape tumors following immunotherapy alone (FC, FCP) also show EMT phenotypes, immunotherapy resistance, and *Irf6* downregulation (see **Response Figure 2a-d** presented in our response to Reviewer #1, above), suggesting that chemotherapy is not required to generate mesenchymal escape variants. We have included **Response Fig. 2a-c** in **Supplementary Fig. 2d-f** of the revised manuscript.

3. In general, the exact treatments and treatment frequencies used in vivo need to be made clear in text and figures. (A) Were mice treated continuously with chemotherapy and immunotherapy regimens? At which timepoints did they receive therapy? This should be marked in the figures. (B) Which combination therapy was used in Fig. 1e-g? (C) What type of ICB did lung cancer patients in Fig. 5e receive? (D) What immunotherapy was given in ED Figs. 5b,c?

Response: We regret the lack of detail. The revision has been updated to include this information.

4. The mechanisms of resistance were studied in escaper cell lines derived from resistant tumors rather than in the original tumor tissue itself. The authors should stain primary tumor tissues harvested from escaper tumors for EMT markers and IRF6 expression, as well as immune markers for different myeloid and lymphocyte populations to confirm the same resistance mechanisms in situ. This should be performed on the 8 escaper samples from which they derived cell lines to also determine the frequency of these EMT and IRF6-driven mechanisms of resistance compared to other mechanisms.

Response: We thank Reviewer #2 for this suggestion. We stained EMT markers in the original primary tumor tissues and observed results similar to those in transplanted tumors, with an obvious decrease in E-cadherin and increases in Vimentin and Twist in Esc tumors. Consistently, almost all Esc lines (Esc 5 was lost) downregulated E-cad and *Irf6* transcripts as well as upregulated EMT-related gene transcripts compared to parental and early progressor (EP) lines (**Response Figure 7a,b**). In addition, we performed immunostaining for IRF6 in the original tumor tissues and found that control

and EP tumors maintained IRF6 expression, whereas all Esc tumors lost IRF6 expression (**Response Figure 7c**). These findings suggest that the acquired resistance to immunotherapy via EMT and IRF6 loss is a widespread phenomenon and not a result of tissue culture passaging. We have included representative tissue staining images in **Supplementary Fig. 2b** and **Figure 4g** in the revised manuscript.

Response Figure 7. EMT/IRF6 loss occurs consistently across Esc tumors. a, Representative IHC images of E-cadherin, Vimentin, and Twist in the original primary tumor tissues obtained from 4662 parental tumor-bearing mice with control IgG (Control) and the relapsed under combination therapy (Esc). **b,** qPCR of E-cad, EMT-related genes, and *Irf6* in 4662 parental, EP, and Esc tumor cells. **c,** Representative IHC images of IRF6 in the original primary tumor tissues from control, EP, and Esc tumor-bearing mice. Scale bars, 250 μ m.

Furthermore, we stained immune markers for myeloid cells (Ly6G) and lymphocytes (CD8a) in the original tumor tissues. While control and EP tumors had abundant infiltrates of Ly6G⁺ cells, almost all Esc tumors contained few Ly6G⁺ cells. By contrast, Esc tumors exhibited greater infiltration of CD8a⁺ cells compared to control and EP tumors (**Response Figure 8a,b**), suggesting that the TME composition of transplanted tumor cell lines was preserved in the original primary tumors. We have included representative tissue staining images and quantifications in the revised manuscript (**Supplementary Fig. 4a,b**).

While we cannot pinpoint the frequency with which acquired resistance is due to EMT-related plasticity, the fact that resistant tumors in our system shared this same phenotype indicates that it is not a unique or rare phenomenon.

Response Figure 8. Original Esc tumors exhibit “hot” tumor phenotypes.

Representative IHC images of Ly6G (**a**) and CD8a (**b**) in the original primary tumor tissues from control, EP, and Esc tumor-bearing mice. Mean counts of positive cells in 2-5 magnification fields per tissue section are shown (right). Scale bars, 250 μ m.

5. Though the authors generated 4 Ctrl, 8 ESC, and 2 EP lines following treatment, it is unclear how many of these lines were used in subsequent in vitro and transplantation experiments. For example, in Fig.1e, is only 1 cell line each for Ctrl, ESC, and EP used, or multiple different cell lines? Importantly, if there were 8 ESC lines generated, did all of them show the same upregulation of EMT markers and phenotypes, suppression of IRF6, and resistance to T cell killing and TNF-induced cell death? How consistent and frequent is this mechanism of resistance? In addition, expression of EMT markers and IRF6, and resistance to T cell killing and TNF-induced cell death should also be interrogated in EP lines as well.

Response: These additional details have been included in the revision. Briefly, **Fig. 1e** shows the combined results of all 4 Ctrl, 8 Esc, and 2 EP lines with 5 or 6 replicates per line. Subsequent gain-of-function and loss-of-function studies represent data with two clonal 4662 and their derived Esc lines. These details are noted in the main text and figure legends. As demonstrated above, Esc lines consistently upregulated the transcripts for EMT markers, suppressed IRF6 expression; likewise, the original tumors shared a poorly differentiated morphology (**Response Figure 7b,c**). Moreover, although Esc lines varied in the expression of MHC I, all Esc lines consistently showed resistance to T-cell killing and TNF-induced cell death (**Response Figure 9a-d**). The characteristics of EP cells were similar to those of parental cells. We have replaced the data in the original manuscript with the more comprehensive data set in revised **Supplementary Fig. 7a,b**.

Response Figure 9. Esc cell lines retain resistance to T cell killing and TNF-induced cell death regardless of Ag:MHC I expression. **a**, Flow cytometry of H-2Kb–OVA₂₅₇₋₂₆₄ in OVA-tdTomato⁺ 4662 parental, EP, and Esc cell lines treated with IFN- γ (100 ng/ml) for 1 d. **b**, AnnexinV and 7-AAD in OVA-tdTomato⁺ 4662 parental, EP, and Esc cell lines co-cultured with activated OT-I for 2 d. **c**, The percentages of 7-AAD⁺ cells among OVA-tdTomato⁺ tumor cells co-cultured as in **b**. **d**, Normalized viability of 4662 parental, EP, and Esc cell lines with varying concentrations of TNF- α in the presence of IFN- γ (0.2 μ g/ml) plus cycloheximide (1 μ g/ml) for 48 h.

6. While experiments were performed in ED Fig. 1e-f to try to rule out the possibility that pre-existing resistant clones may give rise to acquired resistance, since the authors did not use an autochthonous model where heterogeneous epithelial and EMT tumor cell populations spontaneously arise, or transplant KPC cell lines with intermixed epithelial and EMT populations, they cannot fully rule out the possibility that existing clones can give rise to *de novo* resistance to immunotherapy, and should note this in the text and discussion.

Response: We are a bit confused by the Reviewer's point. The results in the original ED **Figure 1e-f** provide strong evidence that resistance arises *de novo* via plasticity, rather than outgrowth of a resistant subclone, since single cell clones still gave rise to a spectrum of therapeutic responses. The Reviewer's comment seems to confirm this interpretation. We have done our best in the Discussion to clarify.

7. Do escaper tumors acquire additional genetic alterations or copy number changes that give rise to immunotherapy resistance? The authors should perform whole exome sequencing and copy number analysis of tissues or cell lines from escapers to rule out

the possibility that acquisition of new genetic alterations could also give rise to resistance.

Response: As suggested by Reviewer #2, we performed copy number analysis comparing escape vs. parental cell lines, as we have done previously (Baslan et al., 2012; Baslan et al., 2015). This analysis revealed *Myc* amplifications, but no other reproducible genomic event, in approximately half of the Esc lines (**Response Figure 10a,b**). As MYC has pleiotropic effects on cell growth, metastasis, EMT, and other features of cancer cell biology, the extent to which *Myc* amplification contributes to the acquired resistance phenotype in our model (as opposed to a more general role in tumor progression, cell line establishment and propagation, etc.) is difficult to ascertain. This will be an interesting line of investigation to pursue in the future, where it will entail substantial functional work. Because of uncertainty regarding the significance of the finding at present, we have elected not to include these data in the revision.

Response Figure 10. *Myc* amplifications are acquired in some, but not all, Esc lines. **a**, Whole genome copy number rendering in integrated genomics viewer of Esc lines E1-E8. Red coloring denotes chromosomal regions that are gained. Blue denotes chromosomal regions that are lost. Black box indicates chromosome 15, where *Myc* is encoded in the mouse genome **b**, Zoom-in-view of chromosome 15 interval that encodes *Myc*. Encoded genes within amplified intervals (e.g. dark red coloring) are labeled.

8. The analysis of T cell priming and activation in the setting of escaper tumors or tumors expressing EMT TFs is insufficient. First, in Fig. 3a-b and ED Fig. 3a-b, total T cell numbers are shown, but T cell activation markers (Ki67, CD69, CD44) and effector functions (GZMB, CD107a, TNFa, IFN γ) are not accessed. It will be critical to know if

T cells are activated in these settings. Moreover, as the authors show that escaper and Zeb1/Snail OE lines are resistant to TNF-induced cell death, it will be important to know if T cells (as well as NK cells) also produce less TNF in these resistance settings. It is possible that resistance could result from a combination of both less T cell activation and secretion of TNF as well as tumor intrinsic TNF insensitivity, which the authors will need to interrogate. Immunohistochemical analyzes of T cell numbers, activation markers, and TNF production will also be informative in primary tumor sections from which these escaper lines were derived. As FCP treatment should increase T cell priming, T cell activation markers and TNF production should be assessed following FCP immunotherapy in mice with parental and escaper tumors.

Response: We thank Reviewer #2 for these valuable suggestions. As suggested, we analyzed the activation and effector molecule expression of cytotoxic T cells infiltrating parental vs. Esc and EV vs. Zeb1/Snail OE tumors upon immunotherapy (FCP). Because we demonstrated that overcoming immune surveillance by T cells, not by NK cells, is important for tumor relapse (**Figure 3d,e**), we focused on the analysis of T cells, especially CD8 T cells. We observed that CD8 T cells infiltrating Esc and Zeb1/Snail OE tumors were activated (Ki67, CD44, CD69) and functionally competent (CD107a, IFN γ , TNF α , Granzyme B, Perforin) compared to those infiltrating parental and EV control tumors, respectively (see **Response Figure 5c,d** provided in response to Reviewer #1, above). We have also now shown increased CD8 T cell infiltrates in the original Esc tumors by IHC (**Response Figure 8b**). Therefore, we have clearly demonstrated that resistance is mainly derived from tumor cell-intrinsic properties rather than T cell inactivation *in vivo*. We have included these data in **Supplementary Fig. 6f,g** of the revision.

9. Though experiments in Fig. 2 and 3 clearly demonstrate that EMT plasticity leads to resistance to immunotherapy and T cell killing, data in Fig. 5c-d show that IRF6 overexpression only has a marginal effect on increasing FCP sensitivity in escaper lines, with only 2/10 mice having CRs. Moreover, only 3/7 lung cancer patients that had acquired resistance to immunotherapy had a decrease in IRF6 expression, suggesting in patients other resistance mechanisms may be more predominant. This questions how important and significant IRF6 loss is to immunotherapy resistance mediated by EMT. Perhaps performing *in vivo* experiments using parental lines with IRF6 knockout may show a more profound effect of the role of IRF6 loss in immunotherapy resistance and are worth pursuing. Given that IRF6 OE seems to increase epithelial markers and decrease mesenchymal markers (ED Fig.7c), and its documented role in regulating E-cadherin (Antiguas et al J Invest Dermatol 2022) and Zeb1 (Li et al Front Oncol 2019), the authors should acknowledge that the impact of IRF6 on immunotherapy outcomes and T cell sensitivity may be in part due to its regulation of the EMT state.

Response: We appreciate the Reviewer's comment and suggestion. We do not claim the EMT/IRF6 mechanism we have uncovered is the sole or even dominant mechanism of acquired resistance to immunotherapy, but only one of several. (It is widely accepted that tumor cells employ multiple mechanisms to evade therapy, with each mechanism reflecting an important biological principle). If anything, our findings

in NSCLC patients (with 3/7 resistant patients having an EMT/IRF6 expression pattern consistent with our murine PDAC results) suggest that this is not a rare phenomenon and may extend across tumor types.

As suggested, we performed experiments in which IRF6 is knocked out in parental tumors. We found that mice bearing IRF6 knockout tumors showed poor response to immunotherapy (FCP) and worse overall survival (**Response Figure 11a,b**). These data have been included in **Figure 5e,f** of the revision.

Response Figure 11. *Irf6* deletion induces resistance to immunotherapy.

Tumor growth (**a**) and survival (**b**) of mice bearing 4662 parental EV and parental *Irf6* KO tumors treated with control IgG or immunotherapy (FCP) ($n = 8$).

Our manuscript does not make any claims about the mechanism of IRF6 activity. It is possible that IRF6 acts in part by participating in or reversing the EMT state, eventually making tumor cells sensitive to cell death stimuli including TNF. However, this is a point of speculation and has not been included.

10. Data in Fig. 6 nicely demonstrates that *Irf6* re-expression sensitizes escaper cells to TNF-induced cell death and T cell killing. Since IRF6 re-expression alone has minimal effects on tumor responses or survival following immunotherapy in vivo (Fig. 5c,d), could activating TNF-induced cell death through treatment with cIAP inhibitors or other small molecules be a more potent and clinically relevant approach to overcome immunotherapy resistance? Similarly, does TNF α blockade in mice with transplanted parental KPC cells lead to resistance to FCP? These experiments would further demonstrate the importance of sensitivity to TNF-induced cell death in EMT-mediated acquired resistance and test a strategy to overcome immunotherapy resistance in PDAC.

Response: These are excellent suggestions, and we performed a series of experiments evaluating cIAP inhibitors and BH3 mimetics to determine whether modulating cell death pathways sensitizes resistant cancer cells to T cell killing. Indeed, we found that the addition of cIAP inhibitor birinapant to T cell co-culture increased sensitivity of both parental and Esc tumor cells to T cell killing cells (**Response Figure 12a**). By contrast, the BH3 mimetics navitoclax and venetoclax induced tumor death on their own but minimally enhanced T cell killing of Esc tumor cells (**Response Figure 12b**).

Response Figure 12. Effects of modulating apoptosis pathways on T cell killing and response to immunotherapy. **a,b**, The percentages of 7-AAD⁺ cells among OVA-tdTomato⁺ 4662 parental and Esc cells co-cultured with or without activated OT-I in the presence of vehicle or birinapant (5 μ M) (**a**) or navitoclax or venetoclax (2 μ M each) (**b**) for 2 d. **c,d**, Tumor growth (**c**) and survival (**d**) of mice bearing 4662 parental EV and *Casp8* KO tumors treated with control IgG or immunotherapy (FCP) ($n = 10$).

Secondly, we tested whether blocking TNF-induced cell death leads to immunotherapy resistance. Because TNF α blockade treatment could induce undesired effects on immune cells, we instead genetically ablated *Casp8*, a central mediator of TNF-induced cell death, in parental cells and assessed immunotherapy responses. Loss of *Casp8* in PDAC cells resulted in marked resistance to immunotherapy and worse mouse survival (**Response Figure 12c,d**). These results suggest that targeting programmed cell death pathways in cancer cells may be a promising strategy to overcome immunotherapy resistance in PDAC, as proposed by the reviewer. We have included **Response Figure 12a,c,d** in the revised manuscript (**Figure 6h-j**).

Minor Comments:

1. Which cell line was transplanted in Fig. 3d? This is not stated.

Response: 4662 parental cells were transplanted. This has been corrected.

2. TPM needs to be defined.

Response: Raw counts of gene transcripts per million (TPM) were generated by Salmon following quasi-alignment of RNA-seq data to the mm39 reference genome.

This has been clarified in the Methods section.

3. Need to show IF staining images for Fig. 6c

Response: We have provided *in vitro* flow cytometry of cleaved caspase-3 (CC3) and *in vivo* CC3 IF staining images (**Response Figure 13a,b**) in the revised manuscript (**Supplementary Figure 11a,b**).

Response Figure 13. *Irf6* re-expression increases cleaved caspase-3 in Esc cells upon pro-apoptotic stimulation. **a**, Flow cytometry of active caspase-3 in 4662 parental EV, Esc EV, and Esc *Irf6* tumors treated with or without TNF- α (0.5 μ g/ml) plus IFN- γ (0.2 μ g/ml) in the presence of cycloheximide (1 μ g/ml) for 48 h. **b**, Representative IF staining images of cleaved caspase-3 in 4662 parental EV, Esc EV, and Esc *Irf6* tumors a week post control IgG or immunotherapy (FCP). Scale bars, 250 μ m.

Reviewer #3 (Remarks to the Author): with expertise in PDAC, immunology, epigenetics, EMT

In this study, Kim and colleagues analyzed the acquired resistance to immunotherapy. Briefly, the authors generated acquired resistance models for GAFCP, reporting that EMT-transcription factors Zeb1 and Snail modulated the tumor cell intrinsic resistance to T cell killing by *Irf6*. Overall, this is an interesting concept, and the findings are potentially important and helpful to advance our current knowledge of pancreatic cancer treatment.

Overall, the quality of the work is high and most of the conclusions are supported by the results. However, the concept/phenomenon that epithelial-mesenchymal plasticity determines immune escape and resistance to immunotherapy in cancer has been reported in several studies and reviews, even though the authors emphasize the acquired resistance. In terms of writing, some results are lack of detailed description, which make it really hard to follow. And some descriptions are overstated.

Response: We thank Reviewer #3 for the positive feedback. We would emphasize that although the relationship between epithelial-mesenchymal plasticity and tumor

immunology has been explored previously, the mechanisms of immune evasion revealed by our study (tumor cell intrinsic features vs. changes in the tumor microenvironment) are quite distinct. We believe this reflects broader differences in the immune evasive mechanisms employed in the setting of long-term (acquired resistance) vs. short-term (primary resistance) immune pressure.

Major points:

1. P5 line 84 “Antitumor responses were associated with immunological memory that was dependent upon T cells but not NK cells (Extended Data Fig. 1b–d). Thus, the combination of CD40 agonist and ICB leads to durable CRs and T cell memory in PDAC”. Flow cytometric analysis of T cells should be included to support this conclusion. The treatment timeline should be included in the legend. This also applies to other similar figures.

Response: We appreciate the Reviewer’s suggestion. To address this point, we performed a set of experiments analyzing the phenotype of T cells (naïve/central memory/effector memory) residing in lymphoid organs in this treatment setting by staining for CD44 and CD62L. We also prepared age-matched naïve and untreated tumor-challenged mice for comparisons. As a result, tumor challenge increased slightly the proportion of effector memory T cells (T_{EM}) in the spleen and draining lymph nodes (dLN); in mice with complete responses (CRs) following immunotherapy (FCP), T_{EM} populations further expanded whereas T_{Naive} populations shrank (**Response Figure 14a,b**). Subsequent tumor rechallenge and T cell depletion experiments support the generation of T cell memory in mice with CR. We have included **Response Figure 14b** in the revised manuscript (**Supplementary Fig. 1b**). The treatment timeline and starting time points have been included in the main text and figures.

Response Figure 14. Combination therapy leads to T cell memory in mice with complete responses. a,b, The frequencies of T_{Naive} , central memory (T_{CM}), and effector memory T cells (T_{EM}) among CD8 T cells in the spleen and draining lymph nodes of naïve, tumor-challenged untreated and immunotherapy (FCP)-treated mice with CR (day 60) analyzed by flow cytometry.

2. Since only one cell line 4662 was used in this study, does the “plasticity” rely on mutations of KRAS and TP53 status? Should be discussed.

Response: This is an important question and one that is difficult to answer since all our syngeneic tumor lines are *Kras/Tp53* mutant. The revised Discussion notes that this remains an outstanding question that needs to be addressed in the future.

3. For NK and CD4/8 depletion assays, the efficiency should be included.

Response: We confirmed the depletion of NK and CD4/8 T cells following treatment with depleting Abs by performing flow cytometry on peripheral blood (**Response Figure 15**). NK and CD4/8 T cell depletions were ~99% efficient (99.01% for NK, 98.63% for CD4, and 99.47% for CD8). This is included in the revised Methods section.

Response Figure 15. Efficient depletion of NK and CD4/8 T cells. The efficiencies of NK and CD4/8 T cell depletion following depleting Ab treatment were determined by flow cytometry on peripheral blood.

4. In Figure 2, considering the existence of many EMT-TFs, why did the authors choose Zeb1 and Snail for the follow-up study? The downregulated or upregulated gene sets used for GSEA analysis should be provided as supplementary data (Figure 2d, e)

Response: Our focus on Zeb1 and Snail was based on earlier profiling experiments examining (i) which EMT-TFs were most consistently upregulated across all Esc cell lines and (ii) which EMT-TFs out of a broad panel had the biggest impact on immunotherapy resistance and mouse survival. Zeb1 and Snail were the clear winners. When we transduced each EMT-TF into parental cells, Zeb1, Snai1, and Snai2, but not Twist1 or Zeb2, aggravated immunotherapy response and mouse survival (**Response Figure 16a,b**). We have included these results in **Supplementary Fig. 3a,b** in the manuscript. **Figure 2d,e** and **ED Figure 2c-i** from the original manuscript have been moved to **Supplementary Fig. 3g,h** and **3c-f, i-k**, respectively. The differentially expressed gene sets used for GSEA analyses have been included in **Supplementary Data 1**.

Response Figure 16. Zeb1, Snai1, and Snai2 impede immunotherapy responses. Tumor growth (a) and survival (b) of mice bearing 4662 parental EV or each EMT-TF transduced tumors treated with control IgG or immunotherapy (FCP) ($n = 10$ to 14).

5. The proportion of tumor-infiltrating gMDSCs, DCs and CD8 T cells were remarkably changed in Esc, and by Zeb1/Snail OE or KO. How about the phenotype of myeloid cells and the cytotoxicity of T/NK cells? Moreover, the authors also need to analyze whether and how Esc and Zeb1/Snail OE/KO cells directly influence the trafficking or function/activity of those altered immune cells.

Response: For our revision, we have performed a more comprehensive analysis of changes in the tumor immune microenvironment. In terms of the myeloid compartment, we observed that both Esc and Zeb1/Snail OE tumors exhibited increased infiltration of macrophages. Specifically, the frequencies of Arginase I⁺ macrophages were elevated in both tumors compared to parental and EV tumors, however, most tumor-infiltrating macrophages were MHC II⁺ M1 macrophages with high expression of CD80 and CD86 (see **Response Figure 5a,b** prepared in response to Reviewer #1 above). Moreover, CD8 T cells in Esc and Zeb1/Snail OE tumors showed intact or even greater extent of activation and cytolytic cytokine production than those in parental and EV tumors (**Response Figure 5c,d**). These findings underscore the notion that the tumor cell intrinsic resistance to T cell killing brought about by EMT creates a hyperactivated immune TME in response. We have included these data in **Supplementary Fig. 6d-g** in the manuscript.

To provide a potential mechanism of how Esc and Zeb1/Snail-modulated tumors influence trafficking or function of immune cells, we performed the cytokine/chemokine array in EV vs. Zeb1/Snail OE tumor cells as well as checked the altered transcripts of cytokines/chemokines in both Esc and Zeb1/Snail OE tumor cell lines. As a result, we found that Esc and Zeb1/Snail OE tumor cells express less MDSC recruiting cytokines and chemokines including G-CSF, GM-CSF, IL-1a, and CXCL2 than parental and EV tumor cells, respectively (see **Response Figure 4a-c**), suggesting that tumor cell plasticity/EMT during acquiring resistance rather creates an immune-favorable microenvironment by regulating tumor cell-derived cytokines/chemokines. These data have been included in **Supplementary Fig. 6a-c** in the manuscript.

6. The analysis detail and result of sequencing data (ATAC, RNA and single cell) should be obtained, in which some important part should be explained in the main text, such as sequencing depth, numbers of detected peaks, and/or information of peaks with differential accessibility. More seriously, the quality of ATAC-seq may be challenged, for too few peaks as common sense. In addition, is there any EMT-related genes changed in the ATAC-seq and/or RNA-seq?

Response: We appreciate these comments. For RNA-seq, we generally obtained 20~25 million reads per sample. For the single cell data, we analyzed published datasets (details regarding depth, read counts, etc. contained therein). The analysis details of ATAC-seq including percent of mitochondria reads, sequencing depth, numbers of detected peaks and peaks with differential accessibility are listed below and have been included in the main text and Methods section.

Sample	% MT reads	# Mapped /Paired Reads	# Peaks	# Peaks with differential accessibility
EV.1	5.41	110729271	48075	16582
EV.2	5.40	144417935	46259	
ZS.1	5.71	118539789	57901	37547
ZS.2	7.49	112658155	56396	
EV OT-1.1	4.26	113973134	46823	16255
EV OT-1.2	5.44	112854631	49130	
ZS OT-1.1	5.10	114909853	53027	31806
ZS OT-1.2	5.09	158375795	51797	

We found that many EMT-related genes were transcriptionally altered in Esc and *Zeb1/Snail* OE tumors by RNA-seq. Representative tracks of EMT-related genes with changes in chromatin accessibility are presented in **Response Figure 17**.

Response Figure 17. EMT-related genes with differential chromatin accessibility by *Zeb1/Snail* OE. a-c, Genome browser tracks showing ATAC-seq reads along *Cdh2* (a), *Twist2* (b), and *Cldn4* (c) genes. Differential accessible sites (arrow) by *Zeb1/Snail* OE were shown.

7. In figure 4, more information about *Irf6* signature genes should be provided.

Response: The *Irf6* signature was obtained from top ranked genes by a STAT value following the DESeq2 analysis of *Irf6* re-expressing Esc vs. EV Esc. The list of genes is now provided in **Supplementary Data 2**.

8. Mechanistically, how does *Zeb1/Snail* OE regulate *Irf6* at the epigenetic and transcriptional level? For example, whether *Zeb1/Snail1* directly bond to *Irf6*-encoding sequence and inhibit its transcription; The large change in chromatin accessibility during EMT plasticity could be mediated by EMT-TFs? Or whether the transcriptional repression and accessibility-loss were driven by histone modification or DNA methylation.

Response: This is an interesting question worth pursuing. To this end, we attempted chromatin immunoprecipitation (ChIP) with antibodies to ZEB1 and SNAIL in parental EV and *Zeb1/Snail* OE (ZS) tumors, followed by qPCR analysis using specific primers for putative *Irf6* promoter region and distal exon 6. These experiments demonstrate that both ZEB1 and SNAIL directly bind to the *Irf6* promoter region, not the distal region (**Response Figure 18**). We have not determined whether further inhibition of *Irf6* transcription is mediated by the transcription repressor activity of these EMT-TFs and/or the recruitment of specific histone modifying enzymes (Aghdassi et al., 2012; Ferrari-Amorotti et al., 2013; Peinado et al., 2004). However, we believe that further detailing the molecular mechanism falls outside the scope of current work. We have included ChIP-qPCR data in **Figure 4f** of the revision.

Response Figure 18. ZEB1 and SNAIL directly bind to the promoter region of *Irf6*. ChIP with control IgG or antibodies to ZEB1 and SNAIL in DNA from 4662 parental EV and *Zeb1/Snail* OE (ZS) tumor cells, followed by qPCR quantification in enriched DNA using primers for *Irf6* putative promoter and distal exon 6 regions.

9. In Figure 5, author used an immunotherapy cohort from lung cancer patients instead of PDAC patients, please provide rational or at least describe the reason in the manuscript.

Response: We agree and recognize that NSCLCs as a group have a far more favorable response to immunotherapy. As noted above, a major objective of our work was to understand the mechanisms of acquired resistance (a major clinical problem in lung cancer) and how they may differ from primary resistance. Consequently, any human correlative evaluation of the mechanisms we revealed – cell plasticity and *Irf6* expression – necessitated investigation of the corresponding clinical scenario. We took great efforts to identify any cohorts in which patients with acquired resistance to immunotherapy had accompanying pre- and post-immunotherapy transcriptome data. The NSCLC dataset from Politi and colleagues was the only study in human cancer we were able to find that met these criteria. We view the fact that we observed an overlapping pattern associated with acquired resistance to ICI in this different tumor type, where one could imagine completely orthogonal resistance mechanisms, as supporting rather than weakening our model. We have included this rationale in the revision.

10. Interestingly, agonistic anti-CD40 Ab could bring about a certain percentage of tumor regressions and responses. Could the authors explain or at least hypothesize the relevance between Esc/EMT/Zeb1/Snail1/Irf6 and CD40 axis?

Response: We previously demonstrated that anti-CD40 therapy exerts anti-tumor effects in pancreatic cancer through macrophages and dendritic cells (Beatty et al., 2011; Diamond et al., 2021; Lin et al., 2020). Engagement of anti-CD40 Ab on these cells could induce TNF production, especially from DCs, or indirectly by activating T cells, the interaction of which is known to be crucial for T cell cancer therapy (Marigo et al., 2016). We observed Esc tumors decreased the expression of *Tnfr1*, and this

was reversed by *Irf6* re-expression in the presence of T cells. The regulation of *Tnfr1* expression might explain the altered sensitivity to TNF and efficacy of anti-CD40 based therapy, however, elucidation of a detailed mechanism by which *Zeb1/Snail-Irf6* axis regulates TNF sensitivity needs further investigation.

Minor concerns

1. Figure 1a, the color line should be removed based on my understanding.

Response: The color line(s) reflect the growth properties of the primary tumors from which the control, EP, and Esc cell lines were generated; this has been better explained in the Figure legend.

2. Figure 1d, “Mice with recurrent tumor after CR.....”, here which group(treatment) it means, or author combine all treatments?

Response: Mice with a relapse after CR by any treatments, achieved only when anti-CD40 and either anti-PD-1 and/or anti-CTLA-4 Abs were combined, were used for this experiment. This has been clarified in the revision.

3. P5 line 82 “However, combination regimens that included agonistic anti-CD40 Ab (F) resulted in tumor regressions and prolonged survival, including complete responses (CRs)” the description of result is not accurate.

Response: We have edited this sentence for accuracy.

4. Too many experimental data were showed only by box graph or dot plot (i.e. Fig 3a-b, 3g-j, S3a-b, 6b-d). The representative flow graph and histopathological figures would improve the credibility.

Response: We have included some representative flow graphs and histopathological images in Supplementary Figures.

5. I highly suggest the authors pay more attention on the English writing to avoid the overstatement. There are some unclear descriptions in Result section, Figure legend, and especially Method, such as Fig 5f-g. Logic and grammar also need to be improved.

Response: We have done our best to edit the manuscript for grammar, logic, and avoidance of hyperbole.

Reviewer #4 (Remarks to the Author): with expertise in PDAC, cancer immunology/immunotherapy

This manuscript describes a novel mechanism of immunotherapy resistance whereby EMT-associated transcription factors silence IRF6, resulting in tumor intrinsic insensitivity to TNF-alpha and T cell mediated killing. They use as a model pre-clinical therapy with agonistic CD40 Ab combinations, which are relevant to the disease. They also employ a series of PDAC models involving cell lines derived from mice differential response to immunotherapy. They later deduced from GSEA that Zeb1 and Snail

overexpression were linked with this escape from immunotherapy efficacy. A variety of mechanistic studies were also performed whereby OT-1 T cells were used to define a T cell defect that was indeed tumor intrinsic. There is also some data from lung cancer patients establishing EMT signatures are enriched in those who do not benefit from immunotherapy to further support their hypothesis. Overall, the manuscript is well written and adds to the field. The data included are quite rigorous with evidence of convincing group sizes and reproducibility. There are a few suggestions below:

Response: We appreciate the positive feedback.

1. It is unclear why the data from orthotopic PDAC models is in the supplement as opposed to the main body of the manuscript. This is a better model and would likely be more of interest to the field if it is more prominently featured in the manuscript. The authors should consider moving it.

Response: As suggested, we have included orthotopic tumor data in the main body of the manuscript (**Figure 2h-l**) and have also left some orthotopic data in supplement (**Supplementary Figure 5a,b**) in the interests of flow and readability.

2. There is no mention of the increase of CD4+ T cells in Figure 3b of the results. This did not seem to happen in the orthotopic model (in supplement). Was it an artifact, or how do the authors explain this result?

Response: **Figure 3b** address changes in CD4+ T cell infiltration; namely, *Zeb1/Snail* OE tumors have increased the infiltration of CD4+ T cells, effector CD4+ T cells as well as regulatory T cells; the latter has been previously reported in association with Snail expression (Kudo-Saito et al., 2009). Orthotopic tumors are generally known to constitute a more immunosuppressive environment compared to subcutaneous tumors. Immunosuppressive cells could reciprocally inhibit T cell infiltration including effector CD4+ T cells into orthotopic tumors.

3. In Figure Legend 1, line 870 is it a typo and should there be an “A” in the “GFCP” abbreviation? Or was abraxane not given?

Response: In this experiment, abraxane was not given based on our finding that abraxane did not have additive effect on GFCP treatment (**Supplementary Fig. 1a**) as well as repeated treatment with Abraxane in mice can cause a lethal hypersensitivity reaction.

4. Discussion, the manuscript is focused primarily on TNF- α mediated killing but other mechanisms of T cell mediated cell death may be operative. Can the authors comment on the role of Perforin and/or granzyme as alternatives and if they suspect IRF6 may influence these other modes of cell killing?

Response: We appreciate this valuable suggestion and agree that it would be useful to examine the role of other mechanisms of T cell-mediated cell death, including perforin and granzyme B. While this is difficult to assess functionally, as there are no specific inhibitors that can be employed, one approach involves the endogenous molecule SerpinB9, which blocks granzyme activity (Jiang et al., 2020). Thus, we

examined the effect of EMT/IRF6 on *Serpib9* expression and found that IRF6 had no significant effect on *Serpib9*, while *Esc* and *Zeb1/Snail* OE tumors had lower *Serpib9* expression (**Response Figure 19a,b**). These results suggest that EMT/IRF6 induce resistance to T cell killing by mechanisms other than blocked granzyme activity.

Response Figure 19. Serpinb9 expression by EMT and *Irf6*. **a**, TPM of *Serpib9* in 4662 parental vs. *Esc* (left) and 4662 parental EV vs. *Zeb1/Snail* OE tumors (right) co-cultured with OT-I. **b**, TPM of *Serpib9* in 4662 *Esc* EV vs. *Esc Irf6* tumors co-cultured with or without OT-I.

5. There are inherent limitations of using an OVA-based system to study antigen specific responses in the context of tumors. These should be more appropriately acknowledged as a limitation in the discussion, since the results were not validated with a true endogenously arising tumor antigen. Likewise the generalizability of the results from lung cancer patients to patients with pancreatic cancer is not discussed. These two tumor types are quite different and it is understandable how the data fits in this paper, however this is not acknowledged.

Response: We appreciate these constructive comments. The limitations of the OVA system, and the rationale for examining datasets from human lung cancer patients (as noted by other Reviewers) are now noted in the revised Discussion.

References (listed alphabetical order by the first author)

- Aghdassi, A., Sendler, M., Guenther, A., Mayerle, J., Behn, C.O., Heidecke, C.D., Friess, H., Buchler, M., Evert, M., Lerch, M.M., and Weiss, F.U. (2012). Recruitment of histone deacetylases HDAC1 and HDAC2 by the transcriptional repressor ZEB1 downregulates E-cadherin expression in pancreatic cancer. *Gut* 61, 439-448.
- Baslan, T., Kendall, J., Rodgers, L., Cox, H., Riggs, M., Stepansky, A., Troge, J., Ravi, K., Esposito, D., Lakshmi, B., *et al.* (2012). Genome-wide copy number analysis of single cells. *Nature protocols* 7, 1024-1041.
- Baslan, T., Kendall, J., Ward, B., Cox, H., Leotta, A., Rodgers, L., Riggs, M., D'Italia, S., Sun, G., Yong, M., *et al.* (2015). Optimizing sparse sequencing of single cells for highly multiplex copy number profiling. *Genome research* 25, 714-724.
- Bayne, L.J., Beatty, G.L., Jhala, N., Clark, C.E., Rhim, A.D., Stanger, B.Z., and Vonderheide, R.H. (2012). Tumor-derived granulocyte-macrophage colony-stimulating factor regulates myeloid inflammation and T cell immunity in pancreatic cancer. *Cancer cell* 21, 822-835.
- Beatty, G.L., Chiorean, E.G., Fishman, M.P., Saboury, B., Teitelbaum, U.R., Sun, W., Huhn, R.D., Song, W., Li, D., Sharp, L.L., *et al.* (2011). CD40 agonists alter tumor stroma and show efficacy against pancreatic carcinoma in mice and humans. *Science* 331, 1612-1616.
- Beatty, G.L., Winograd, R., Evans, R.A., Long, K.B., Luque, S.L., Lee, J.W., Clendenin, C., Gladney, W.L., Knoblock, D.M., Guirnalda, P.D., and Vonderheide, R.H. (2015). Exclusion of T Cells From Pancreatic Carcinomas in Mice Is Regulated by Ly6C(low) F4/80(+) Extratumoral Macrophages. *Gastroenterology* 149, 201-210.
- Diamond, M.S., Lin, J.H., and Vonderheide, R.H. (2021). Site-Dependent Immune Escape Due to Impaired Dendritic Cell Cross-Priming. *Cancer immunology research* 9, 877-890.
- Ferrari-Amorotti, G., Fragiasso, V., Esteki, R., Prudente, Z., Soliera, A.R., Cattelani, S., Manzotti, G., Grisendi, G., Dominici, M., Pieraccioli, M., *et al.* (2013). Inhibiting interactions of lysine demethylase LSD1 with snail/slug blocks cancer cell invasion. *Cancer research* 73, 235-245.
- Jiang, L., Wang, Y.J., Zhao, J., Uehara, M., Hou, Q., Kasinath, V., Ichimura, T., Banouni, N., Dai, L., Li, X., *et al.* (2020). Direct Tumor Killing and Immunotherapy through Anti-SerpineB9 Therapy. *Cell* 183, 1219-1233 e1218.
- Kudo-Saito, C., Shirako, H., Takeuchi, T., and Kawakami, Y. (2009). Cancer metastasis is accelerated through immunosuppression during Snail-induced EMT of cancer cells. *Cancer cell* 15, 195-206.
- Li, J., Byrne, K.T., Yan, F., Yamazoe, T., Chen, Z., Baslan, T., Richman, L.P., Lin, J.H., Sun, Y.H., Rech, A.J., *et al.* (2018). Tumor Cell-Intrinsic Factors Underlie Heterogeneity of Immune Cell Infiltration and Response to Immunotherapy. *Immunity* 49, 178-193 e177.
- Lin, J.H., Huffman, A.P., Wattenberg, M.M., Walter, D.M., Carpenter, E.L., Feldser,

D.M., Beatty, G.L., Furth, E.E., and Vonderheide, R.H. (2020). Type 1 conventional dendritic cells are systemically dysregulated early in pancreatic carcinogenesis. *The Journal of experimental medicine* 217.

Marigo, I., Zilio, S., Desantis, G., Mlecnik, B., Agnellini, A.H.R., Ugel, S., Sasso, M.S., Qualls, J.E., Kratochvill, F., Zanovello, P., *et al.* (2016). T Cell Cancer Therapy Requires CD40-CD40L Activation of Tumor Necrosis Factor and Inducible Nitric-Oxide-Synthase-Producing Dendritic Cells. *Cancer cell* 30, 377-390.

Markosyan, N., Li, J., Sun, Y.H., Richman, L.P., Lin, J.H., Yan, F., Quinones, L., Sela, Y., Yamazoe, T., Gordon, N., *et al.* (2019). Tumor cell-intrinsic EPHA2 suppresses anti-tumor immunity by regulating PTGS2 (COX-2). *The Journal of clinical investigation* 129, 3594-3609.

O'Hara, M.H., O'Reilly, E.M., Varadhachary, G., Wolff, R.A., Wainberg, Z.A., Ko, A.H., Fisher, G., Rahma, O., Lyman, J.P., Cabanski, C.R., *et al.* (2021). CD40 agonistic monoclonal antibody APX005M (sotigalimab) and chemotherapy, with or without nivolumab, for the treatment of metastatic pancreatic adenocarcinoma: an open-label, multicentre, phase 1b study. *The Lancet. Oncology* 22, 118-131.

Padron, L.J., Maurer, D.M., O'Hara, M.H., O'Reilly, E.M., Wolff, R.A., Wainberg, Z.A., Ko, A.H., Fisher, G., Rahma, O., Lyman, J.P., *et al.* (2022). Sotigalimab and/or nivolumab with chemotherapy in first-line metastatic pancreatic cancer: clinical and immunologic analyses from the randomized phase 2 PRINCE trial. *Nature medicine* 28, 1167-1177.

Peinado, H., Ballestar, E., Esteller, M., and Cano, A. (2004). Snail mediates E-cadherin repression by the recruitment of the Sin3A/histone deacetylase 1 (HDAC1)/HDAC2 complex. *Molecular and cellular biology* 24, 306-319.

Winograd, R., Byrne, K.T., Evans, R.A., Odorizzi, P.M., Meyer, A.R., Bajor, D.L., Clendenin, C., Stanger, B.Z., Furth, E.E., Wherry, E.J., and Vonderheide, R.H. (2015). Induction of T-cell Immunity Overcomes Complete Resistance to PD-1 and CTLA-4 Blockade and Improves Survival in Pancreatic Carcinoma. *Cancer immunology research* 3, 399-411.

REVIEWERS' COMMENTS

Reviewer #1 (Remarks to the Author):

The revision has addressed my prior comments. I don't have further comment.

Reviewer #2 (Remarks to the Author):

The authors have done an excellent job addressing our comments, including performing a large number of new experiments. This has helped to substantiate the authors conclusions and significance of their findings, and as such the revised manuscript is much improved.

We have just a few remaining comments on additional data to include in the manuscript and points to make in the Discussion:

1. Response Fig. 6: For the orthotopic transplantation experiments, the authors should include the tumor growth curves showing the kinetics of response and relapse in the manuscript, similar to what is shown for subcutaneous transplant tumors in Fig. 1a.
2. Heterogeneous drug responses to single cell clone transplants in ED Fig. 1f is not sufficient to make the claim that acquired resistance is the product of plasticity rather than the outgrowth of pre-existing mutant subclones. Since cells with EMT properties can be found early during tumor development in KPC GEMM mice as previously published by the investigators (Rhim et al Cell 2012), it is likely that pre-existing clones with EMT features could also contribute to de novo resistance in an autochthonous tumor setting. Since the authors experiments cannot rule out this possibility they should comment on this limitation in their study in the Discussion.
3. The copy number analysis showing MYC amplification in escaper tumors is quite striking (Response Fig. 10). If the authors decide to not include this data in the manuscript, they will need to comment in the Discussion that they cannot rule out the possibility that genetic alterations and copy number changes acquired during immunotherapy resistance might also contribute to relapse in addition to EMT plasticity.

Reviewer #3 (Remarks to the Author):

The author has answered all the questions I had. No further questions.

Reviewer #4 (Remarks to the Author):

The authors have substantially revised the manuscript and addressed my comments. They have done a significant amount of work and new experiments including much more relevant orthotopic tumor models suggested by other reviewers that produced data in alignment with their other results. This manuscript has improved and I have no further comments. This finding will be of interest to the field and advance our understanding of immune suppression in pancreatic cancer.

Point-by-point Response to the Reviewers

RE: MS# NCOMMS-23-23245A

"Plasticity-induced repression of Irf6 underlies acquired resistance to cancer immunotherapy"

Reviewer #1 (Remarks to the Author):

The revision has addressed my prior comments. I don't have further comment.

Response: We appreciate the Reviewer's enthusiasm for the revised manuscript.

Reviewer #2 (Remarks to the Author):

The authors have done an excellent job addressing our comments, including performing a large number of new experiments. This has helped to substantiate the authors conclusions and significance of their findings, and as such the revised manuscript is much improved.

We have just a few remaining comments on additional data to include in the manuscript and points to make in the Discussion:

Response: We thank Reviewer #2 for the positive comments.

1. Response Fig. 6: For the orthotopic transplantation experiments, the authors should include the tumor growth curves showing the kinetics of response and relapse in the manuscript, similar to what is shown for subcutaneous transplant tumors in Fig. 1a.

Response: We appreciate the Reviewer's comment. In order to extend our findings in the context of the native microenvironment, we created a large cohort of mice bearing orthotopic 4662 tumors with immunotherapy (FCP). By ultrasound monitoring, we were able to observe tumor regression in response to immunotherapy and relapses several weeks after definite regression (**Response Figure 1**). Consequently, we found that Esc cell lines from these relapsed tumors exhibit EMT phenotypes in a consistent manner, and successfully recapitulate our findings in the subcutaneous setting. However, the limitation of ultrasound monitoring is that normal tumor-free pancreas is also detected in a mean diameter range of 2-3 mm (we confirmed this in cured mice later). This may give rise to confusion, and for this reason we have chosen not to include this data in our manuscript.

Response Figure 1. Ultrasound monitoring of orthotopic tumor response and recurrence upon immunotherapy.

2. Heterogeneous drug responses to single cell clone transplants in ED Fig. 1f is not sufficient to make the claim that acquired resistance is the product of plasticity rather than the outgrowth of pre-existing mutant subclones. Since cells with EMT properties can be found early during tumor development in KPC GEMM mice as previously published by the investigators (Rhim et al Cell 2012), it is likely that pre-existing clones with EMT features could also contribute to de novo resistance in an autochthonous tumor setting. Since the authors experiments cannot rule out this possibility they should comment on this limitation in their study in the Discussion.

Response: We thank Reviewer #2 for this comment. We have additionally described the limitation of our study and a future direction regarding this point in the Discussion.

3. The copy number analysis showing MYC amplification in escaper tumors is quite striking (Response Fig. 10). If the authors decide to not include this data in the manuscript, they will need to comment in the Discussion that they cannot rule out the possibility that genetic alterations and copy number changes acquired during immunotherapy resistance might also contribute to relapse in addition to EMT plasticity.

Response: We appreciate the Reviewer's point. We have briefly mentioned this observation and noted the potential contribution of genetic or SCNAs to acquired immunotherapy resistance.

Reviewer #3 (Remarks to the Author):

The author has answered all the questions I had. No further questions.

Response: We appreciate the Reviewer's enthusiasm for the revised manuscript.

Reviewer #4 (Remarks to the Author):

The authors have substantially revised the manuscript and addressed my comments. They

have done a significant amount of work and new experiments including much more relevant orthotopic tumor models suggested by other reviewers that produced data in alignment with their other results. This manuscript has improved and I have no further comments. This finding will be of interest to the field and advance our understanding of immune suppression in pancreatic cancer.

Response: We appreciate the Reviewer's enthusiasm for the revised manuscript.